# Conformalized Multiple Testing after Data-dependent Selection

**Xiaoning Wang**[1*]    **Yuyang Huo**[1*]    **Liuhua Peng**[2†]    **Changliang Zou**[1†]

[1]School of Statistics and Data Sciences, LPMC, KLMDASR and LEBPS,
Nankai University, Tianjin, China
[2]School of Mathematics and Statistics, The University of Melbourne, Melbourne, Australia
1120220048@mail.nankai.edu.cn,huoyynk@gmail.com
liuhua.peng@unimelb.edu.au, zoucl@nankai.edu.cn

## Abstract

The task of distinguishing individuals of interest from a vast pool of candidates using predictive models has garnered significant attention in recent years. This task can be framed as a *conformalized multiple testing* procedure, which aims at quantifying prediction uncertainty by controlling the false discovery rate (FDR) via conformal inference. In this paper, we tackle the challenge of conformalized multiple testing after data-dependent selection procedures. To guarantee the construction of valid test statistics that accurately capture the distorted distribution resulting from the selection process, we leverage a holdout labeled set to closely emulate the selective distribution. Our approach involves adaptively picking labeled data to create a calibration set based on the stability of the selection rule. This strategy ensures that the calibration data and the selected test unit are exchangeable, allowing us to develop valid conformal p-values. Implementing with the famous Benjamini-Hochberg (BH) procedure, it effectively controls the FDR over the selected subset. To handle the randomness of the selected subset and the dependence among the constructed p-values, we establish a unified theoretical framework. This framework extends the application of conformalized multiple testing to complex selective settings. Furthermore, we conduct numerical studies to showcase the effectiveness and validity of our procedures across various scenarios.

## 1 Introduction

In recent years, there has been a notable focus on the use of predictive models to distinguish specific individuals from a pool of candidates. For instance, in the field of financial investment [17, 2], machine learning models can be used to predict profits for different investment opportunities. Candidates with high predicted profits can then be given more preference and considered as potential investment options. Similarly, in disease diagnostics [29, 45], researchers can utilize relevant information and corresponding predictions from machine learning models to identify potential patients.

In a typical scenario, we are presented with a labeled/holdout data set $\mathcal{D}_c = \{Z_i = (X_i, Y_i)\}_{i=1}^n$, where $X_i \in \mathbb{R}^d$ is the observed covariate and $Y_i \in \mathbb{R}$ is the response, and an unlabelled/test set $\mathcal{D}_u = \{X_i\}_{i=n+1}^{n+m}$. In practice, we only observe the covariates in the test set $\mathcal{D}_u$ and the responses $Y$ are unknown. Our goal is to distinguish individuals in $\mathcal{D}_u$ whose undisclosed responses fall within a predetermined region $\mathcal{A}$. Region $\mathcal{A}$ can take various forms, such as $(b, \infty)$, $[a, b]$ and $(-\infty, a)$ per user's requirements. To estimate the value of $Y$ for the identification, we employ a predictive model $\hat{\mu} : \mathbb{R}^d \to \mathbb{R}$. However, directly using the black-box prediction $\hat{\mu}(X_i)$ as a substitute for $Y_i$ leads to

---

[*]Equal contribution.
[†]Correspondence to: Liuhua Peng <liuhua.peng@unimelb.edu.au>, Changliang Zou <zoucl@nankai.edu.cn>.

inherent uncertainty. In order to quantify this uncertainty, we reformulate our problem as multiple hypothesis testing [10]: for $j = \{n+1, \cdots, n+m\}$,

$$H_{0,j} : Y_j \notin \mathcal{A} \quad \text{v.s.} \quad H_{1,j} : Y_j \in \mathcal{A}.$$

Under this framework, we wish to make rejections as much as possible with the false discovery rate (FDR) controlled at a pre-given level $\alpha$. Denote the index set of test data as $\mathcal{U} = \{n+1, \cdots, n+m\}$. The FDR is defined as the expectation of false positive proportion (FDP) over the test units in $\mathcal{U}$, i.e.

$$\text{FDR}(\mathcal{U}) = \mathbb{E}[\text{FDP}(\mathcal{U})], \quad \text{FDP}(\mathcal{U}) = \frac{\sum_{j \in \mathcal{U}} \mathbb{1}\{j \in \mathcal{R}, Y_j \notin \mathcal{A}\}}{1 \vee |\mathcal{R}|}.$$

where we denote $a \vee b = \max\{a, b\}$ for any $a, b \in \mathbb{R}$, $|\mathcal{S}|$ as the cardinality of a set $\mathcal{S}$ and $\mathcal{R}$ as the rejection set. To derive the testing rule, one feasible method is to construct the conformal p-value [8] by ranking the nonconformity score associated with the test unit's prediction among the scores in the holdout set. Then we can apply the well-known Benjamini-Hochberg (BH) procedure [10] on these conformal p-values to obtain a rejection set with controlled FDR. Here we term this procedure as "conformalized multiple testing".

In practice, researchers may be interested in specific subsets rather than analyzing the entire dataset. For example, they might aim to determine the presence or absence of lung cancer among heavy smokers. By establishing data-driven criteria or thresholds based on factors such as the daily number of cigarettes smoked and smoking history, researchers can create a filtered subset consisting of heavy smokers. This allows researchers to gain insights into the patterns of the medical condition within this particular group. Here we address that the group partition may not be predetermined and could instead be learned from data, through methods like clustering [25] or thresholding. Denote the selected subset from unlabelled data as $\hat{\mathcal{S}}_u \subset \mathcal{U}$. In our paper, we aim to find a rejection set $\hat{\mathcal{R}}_u \subset \hat{\mathcal{S}}_u$ with the following FDR criterion controlled at $\alpha$, i.e.

$$\text{FDR}(\hat{\mathcal{S}}_u) = \mathbb{E}\left[\frac{\sum_{j \in \hat{\mathcal{S}}_u} \mathbb{1}\{j \in \hat{\mathcal{R}}_u, Y_j \notin \mathcal{A}\}}{1 \vee |\hat{\mathcal{R}}_u|}\right] \leq \alpha.$$

For simplicity, we use FDR to denote $\text{FDR}(\hat{\mathcal{S}}_u)$ only. The selection procedure would distort the distribution of test statistics, invalidating the p-values and leading to the failure of FDR control. This falls into the category of selective inference, which has been addressed in both statistics and machine learning fields [53, 16]. To tackle the selective issue, the use of labeled data becomes crucial. By leveraging labeled data, it is possible to obtain the conditional distribution of the selected data, which in turn allows for the construction of valid p-values.

Nevertheless, even with valid p-values, controlling the FDR proves to be a challenging task. This difficulty arises from the inherent randomness of the selected subset $\hat{\mathcal{S}}_u$. Even a minor disturbance in $\hat{\mathcal{S}}_u$ can lead to significant changes in the final rejection set $\hat{\mathcal{R}}_u$. Consequently, in order to address this issue, we focus on several commonly used selection rules with certain selection stability. We employ an adaptive strategy to carefully choose labeled data, thereby creating a calibration set that takes into account the selection stability.

## 1.1 Our contributions

In this paper, we construct the *selective conformal p-value* for each selected individual, built on the marginal conformal p-value [8]. To address the selection effects, we adaptively pick a calibration set from the labeled data according to the selection rule, to ensure the exchangeability between the test data and labeled data. The selective conformal p-values are then constructed using the picked calibration set. By combining the selective conformal p-values with the well-known BH procedure, we achieve FDR control after data-driven selection, as verified through our comprehensive analysis.

The main contributions of our paper can be summarized as follows.

- Firstly, we frame the problem of multiple testing after data-dependent selection in the predictive setting and propose a viable solution utilizing the labeled data.
- Secondly, the proposed method achieves exact FDR control for selection rules with strong stability, including joint-exchangeable rules and the top-K selection. And we further extend

our method to handle more general cases where the selection rules satisfy a weaker stability condition such as sample mean selection.

- Thirdly, the theoretical advancement extends the scope of classic multiple testing into the selective setting, providing a unified analytical technique for handling the randomness arising from data-driven selection.

- Finally, through extensive experiments, we evaluate the reliability of our method in delivering the desired FDR control, and emphasize its easy integration with various algorithms.

## 1.2 Connections to existing works

**Multiple testing** Ever since the seminal work of Benjamini and Hochberg [10], the framework of multiple testing has been well developed by many researchers [49, 5, 66, 20, 44]. Our paper connects to the area of two-stage testing [62], which firstly selects a subset of hypotheses and subsequently applies a multiple testing procedure to the selected set. To maintain the validity of test statistics after the selection, Bourgon et al. [14] recommended using an independent statistic specifically for the purpose of selection, but it is unavailable in our predictive setting. Instead of assuming independence between the test statistic and the selection statistic, Du and Zhang [19] introduced the concept of a "single-index" p-value for the joint modeling of both statistics and provided FDR control under symmetry assumption. Besides, Efron [21] considers applying multiple testing procedures over pre-given groups to guarantee group-wise FDR control, while our work considers the FDR control over data-driven subgroups, which is more challenging.

**Conformal inference** Conformal inference [57, 40, 56, 55] has garnered significant attention in recent years, which leverages data exchangeability to construct model-agnostic prediction intervals. We present some recent developments therein [39, 6, 59, 61, 50, 18, 15, 1]. Within the conformal inference framework, several studies have focused on controlling the FDR in predictive setting, i.e. the *conformalized multiple testing* [8, 26]. These studies involve constructing a valid testing procedure using a holdout set. Such procedures include the BH procedure based on conformal p-values [27, 35, 34, 24], thresholding via an FDP estimator [63, 41] and the e-BH procedure applied to generalized e-values [7, 64]. Different from them, we focus on a selective scenario in conformalized multiple testing, addressing new challenges arising from selective randomness.

**Selective inference** Selective inference concerns the inference problem after data-dependent processing. Previous works have mainly focused on the inference of parameters [60, 31]. Recently, Bao et al. [3] extended selective inference to the realm of conformal inference. They proposed a method to construct selection conditional prediction intervals with controlled false coverage-statement rate (FCR) [12] after data selection. Building upon this work, selective conformal inference with FCR control has been further extended to accommodate more general selection rules [23, 28] or the online setting [4]. In particular, under a certain class of selection rules, Gazin et al. [23] involves a procedure for FDR control after selection which closely aligns with our method. However, their approach focuses on selecting an informative set with FCR control under specific selection assumptions, limiting its applicability in more general scenarios such as those with data-dependent selection. The problem we tackle presents additional complexities, due to the intricate dependence on the selection procedure and final decision procedure, requiring a more intricate and delicate analysis. Besides, Sarkar and Kuchibhotla [47] proposed a post-selection framework to guarantee simultaneous inference [13] for all coverage levels, which differs greatly from our scenarios.

## 2 Methodology

### 2.1 Recap: conformalized multiple testing

We first introduce how to make multiple testing in the predictive setting. Denote the index sets for the labelled data $\mathcal{D}_c$ as $\mathcal{C} = \{1, \cdots, n\}$. Suppose $\hat{\mu}(\cdot)$ is a predictor via a machine learning algorithm that is pre-given or can be trained on extra labeled data. Thus we can treat $\hat{\mu}(x)$ as fixed. To construct a valid test statistic based on $\hat{\mu}(X_j)$, Bates et al. [8] considered to use conformal p-values built upon the conformal inference framework [56]. Consider a monotone transformation $V$ such that the larger value of $V(\hat{\mu}(X_j))$ indicates the bigger likelihood of $Y \notin \mathcal{A}$. For example, if $\mathcal{A} = (b, \infty)$, we can

use $V(y) = b - y$. The marginal conformal p-value $p_j^{\mathrm{M}}$ for $X_j$ is defined as

$$p_j^{\mathrm{M}} = \frac{1 + |\{i \in \mathcal{C}_0 : V_i \le V_j\}|}{1 + |\mathcal{C}_0|}, \quad j \in \mathcal{U}; \tag{1}$$

where we denote $V_j = V(\hat{\mu}(X_j))$ as the nonconformity score for $j$-th sample and $\mathcal{C}_0 = \{i \in \mathcal{C} : Y_i \notin \mathcal{A}\}$ as the index set of labeled set containing only null samples.

The properties of marginal conformal p-value constructed by i.i.d.~labeled and test data have been investigated by Bates et al. [8] and we present them in the following proposition. Proposition 2.1 (i) guarantees that marginal conformal p-value is superuniform, thus it is a valid p-value. As the conformal p-values have a nice dependence structure, the rejection set obtained by the famous BH procedure [10] enjoys valid FDR control Proposition 2.1 (ii) indicates. For a set of p-values $\{p_i\}_{i=1}^m$, the BH procedure finds $k = \max\{j : p_{(j)} \le j\alpha/m\}$ where $p_{(j)}$ denotes the $j$-th smallest p-value in $\{p_j\}_{j=1}^m$ and obtain the rejection set as $\hat{\mathcal{R}}_u = \{j : p_j \le k\alpha/m\}$.

**Proposition 2.1** (Properties of the conformal p-value [8]). *Suppose the labeled data and test data are i.i.d.. For simplicity, we assume $\mathcal{U}_0 = \{j \in \mathcal{U} : Y_j \notin \mathcal{A}\} = \{n+1, \cdots, n+m_0\}$ for $m_0 \le m$. The conformal p-values in* (1) *satisfy:*

*(i) The $p_j^{\mathrm{M}}$ is a marginally superuniform p-value, i.e. for any $t$, $\Pr(p_j^{\mathrm{M}} \le t \mid j \in \mathcal{U}_0) \le t$.*

*(ii) Furthermore, the BH procedure applied at level $\alpha$ on the conformal p-values $\{p_j^{\mathrm{M}}\}_{j \in \mathcal{U}}$ controls the FDR level at $\pi\alpha$, where $\pi$ is the null proportion of test samples.*

The property of the conformal p-value is obtained by the exchangeability between the labeled data and test data. Through this, we can approximate the distribution of $V_j$, where $j \in \mathcal{U}_0$, using $V_i$ from $i \in \mathcal{C}_0$. Therefore, when the labeled data and test data have different distributions, maintaining the exchangeability becomes crucial in constructing valid conformal p-values.

As Storey [48] suggested, we can further estimate the null proportion and incorporate it in the BH procedure to further increase detection power. With the aid of labeled data, the null proportion can be directly estimated by the corresponding proportion in the labeled set, i.e. $\hat{\pi} = |\mathcal{C}_0|/|\mathcal{C}|$.

## 2.2 Selective conformal p-value

A selection procedure could possibly be employed to the test samples. In this case, the focus lies primarily on the selected subgroup rather than the entire dataset, and decisions are made solely based on this subset. Define the selection rule as $\mathbf{S}_{\mathcal{D}_c, \mathcal{D}_u}$, which is a function of labeled set $\mathcal{D}_c$ and test set $\mathcal{D}_u$. For simplicity, we may omit this subscript. The selection rule maps an individual point $X$ into a selection decision $\{0, 1\}$. And the selected subset can be written as $\hat{\mathcal{S}}_u = \{j \in \mathcal{U} : \mathbf{S}(X_j) = 1\}$.

There are many examples for data-dependent $\mathbf{S}$. The $\mathbf{S}$ can be the clustering algorithm which automatically determines the subgroup. Alternatively, $\mathbf{S}$ is associated with the selection score $T_j$, which is derived from certain components of $X_j$, and a selection threshold $\tau$, such as the sample mean value in $\{T_j\}_{j \in \mathcal{U}}$. In this case, we can express $\{\mathbf{S}(X_j) = 1\} = \{T_j < \frac{1}{m}\sum_{k \in \mathcal{U}} T_k\}$.

After the selection procedure, we would make multiple testing on the selected subset $\hat{\mathcal{S}}_u$. However, the distribution of the selected conformal p-values in (1) would be distinct from the original ones, due to the selection effects. Consequently, directly running the BH procedure on the marginal conformal p-values (1) in selected set $\hat{\mathcal{S}}_u$ has no guarantee, which may lead to an inflated FDR level or poor power. Addressing this issue raises two important considerations:

- How to characterize the selection conditional distribution of the selected individuals to construct valid p-values?

- How to take account of the dependence structure among the valid p-values and the stochastic nature of the selection event to design a trustful multiple testing procedure?

The first issue is widely considered in post-selection inference. Previous literature heavily relies on the normality assumption to derive the conditional distribution of test statistics [32, 52]. By the spirit of conformal inference, we consider constructing the selective conformal p-value by picking up the

calibration set via the same selection rule, thereby guaranteeing the exchangeability between the selected test unit and picked calibration data.

To be specific, we employ the selective algorithm $\mathbf{S}$ on the labeled set to derive the picked calibration set $\hat{\mathcal{S}}_c = \{i \in \mathcal{C} : \mathbf{S}(X_i) = 1\}$. If $\mathbf{S}$ involves a selection threshold $\tau$, we will choose the calibration set as $\hat{\mathcal{S}}_c = \{i \in \mathcal{C} : T_i \leq \tau\}$. We hope $\{V_i : i \in \hat{\mathcal{S}}_c\} \cup V_j$ for $j \in \hat{\mathcal{S}}_u$ exhibits a certain level of exchangeability, enabling us to capture the distribution of $V_j$.

With the aid of $\hat{\mathcal{S}}_c$, the selective conformal p-value can be accordingly constructed as:

$$p_j := \frac{1 + |\{i \in \hat{\mathcal{S}}_c \cap \mathcal{C}_0 : V_i \leq V_j\}|}{1 + |\hat{\mathcal{S}}_c \cap \mathcal{C}_0|}, \quad \text{for } j \in \hat{\mathcal{S}}_u. \tag{2}$$

After obtaining valid p-values, ensuring the guarantee of the BH procedure is not straightforward due to the dependence arising from the use of the same calibration set in computing conformal p-values and the randomness from the data-dependent selection. Therefore, the second concern needs to be carefully addressed. In this article, we examine the BH procedure applied to the selective conformal p-values can enjoy finite sample FDR control for several commonly used selection rules. We outline our procedure in Algorithm 1 and refer to our method as Selective Conformal P-Value (SCPV).

---

**Algorithm 1** Selective conformal p-value with BH procedure (SCPV)

---

**Input:** Labeled set $\mathcal{D}_c$, test set $\mathcal{D}_u$, selection procedure $\mathbf{S}_{\mathcal{D}_c, \mathcal{D}_u}$, prediction model $\hat{\mu}(\cdot)$, target FDR level $\alpha \in (0, 1)$.

    **Step 1** (Selection) Apply the selective procedure $\mathbf{S}_{\mathcal{D}_c, \mathcal{D}_u}$ to obtain the selected subsets $\hat{\mathcal{S}}_u$ and $\hat{\mathcal{S}}_c$.

    **Step 2** (Calibration) Compute $\{V_i : i \in \hat{\mathcal{S}}_c \cap \mathcal{C}_0\}$, $\{V_j : j \in \hat{\mathcal{S}}_u\}$.

    **Step 3** (Construction) Construct selective conformal p-value for each $j \in \hat{\mathcal{S}}_u$ as (2)

    **Step 4** (BH procedure) Compute $k^* = \max\{k : \sum_{j \in \hat{\mathcal{S}}_u} \mathbb{1}(p_j \leq \alpha k / m) \geq k\}$

**Output:** Rejection set $\hat{\mathcal{R}}_u = \{j \in \hat{\mathcal{S}}_u : p_j \leq \alpha k^* / m\}$.

---

# 3 Theoretical guarantee

In this section, we aim to verify the FDR guarantee of Algorithm 1 for several commonly used selection rules. Our focus here is to tackle the technical challenges associated with the selective multiple testing problem. Unlike the conventional approach where a fixed number $m$ of test units is considered, we encounter a challenge due to the involvement of a random number of test units $|\hat{\mathcal{S}}_u|$. This randomness makes the analysis considerably intricate unless we impose certain restrictions on the selection rule. To deal with the selection set $\hat{\mathcal{S}}_u$, we introduce the concept of strong stability.

**Definition 3.1** (Strong stability). Given selection set $\hat{\mathcal{S}}_u = \{j \in \mathcal{U} : \mathbf{S}_{\mathcal{D}_c, \mathcal{D}_u}(X_j) = 1\}$. The selection rule $\mathbf{S}_{\mathcal{D}_c, \mathcal{D}_u}$ is strongly stable if either of the conditions holds: for any $i \in \mathcal{C} \cup \mathcal{U}$ and $j \in \hat{\mathcal{S}}_u$

- (Leave out) $\mathbf{S}_{\mathcal{D}_c, \mathcal{D}_u}(X_i) = \mathbf{S}_{\mathcal{D}_c \cup \{Z_j\}, \mathcal{D}_u \setminus \{Z_j\}}(X_i)$;

- (Replace) $\mathbf{S}_{\mathcal{D}_c, \mathcal{D}_u}(X_i) = \mathbf{S}_{\mathcal{D}_c, \mathcal{D}_u \setminus \{Z_j\} \cup \{z\}}(X_i)$ for a fixed point $z$.

Here we define the strong stability of selection rule in two common ways: leaving one point out or replacing one point with a fixed value. Many popular selection rules are strongly stable, such as joint-exchangeable rule and top-K selection. Detailed discussions are provided in next subsections.

The strong stability plays a crucial role in our analysis, as it enables us to fix the randomness of the selected set $\hat{\mathcal{S}}_u$. With the strongly stable property, we can perform a delicate analysis for the rejection set from Algorithm 1 to obtain the theoretical guarantee.

**Theorem 3.2.** *Suppose the data are i.i.d. and the selection rule $\mathbf{S}_{\mathcal{D}_c, \mathcal{D}_u}$ is strongly stable. Then the selective conformal p-values defined in* (2) *satisfies* $\Pr(p_j \leq t | j \in \hat{\mathcal{S}}_u, j \in \mathcal{U}_0) \leq t$, *and the output $\hat{\mathcal{R}}_u$ of Algorithm 1 satisfies* $\mathrm{FDR} \leq \alpha \mathbb{E}[|\hat{\mathcal{S}}_u \cap \mathcal{U}_0| / |\hat{\mathcal{S}}_u|] \leq \alpha$.

We present the insight of our proof for Theorem 3.2. As a common operation in analyzing FDR [22, 35], we decompose the FDR into the FDR contribution for each $j \in \mathcal{U}$ as

$$\text{FDR} = \mathbb{E}\left[\frac{\sum_{j\in\mathcal{U}} \mathbb{1}\{j \in \hat{\mathcal{R}}_u, j \in \mathcal{U}_0\}}{1 \vee |\hat{\mathcal{R}}_u|}\right] = \sum_{j\in\mathcal{U}} \mathbb{E}\left[\frac{\mathbb{1}\{p_j \leq \alpha\frac{|\hat{\mathcal{R}}_u|}{|\hat{\mathcal{S}}_u|}, j \in \hat{\mathcal{S}}_u, j \in \mathcal{U}_0\}}{1 \vee |\hat{\mathcal{R}}_u|}\right].$$

By the stability of selection rule, we can replace $|\hat{\mathcal{R}}_u|$ with a decoupled version $|\hat{\mathcal{R}}_u^{(j)}|$ which removes the influence of $p_j$. If given some quantity $\Phi_j$ that blocks most of the nuisance parameters, the p-value $p_j$ has a uniform distribution and $|\hat{\mathcal{R}}_u^{(j)}|, |\hat{\mathcal{S}}_u|$ are fixed. Then the FDR control is by

$$\text{FDR} = \sum_{j\in\mathcal{U}} \mathbb{E}\left[\frac{\mathbb{E}[\mathbb{1}\{p_j \leq \alpha\frac{|\hat{\mathcal{R}}_u^{(j)}|}{|\hat{\mathcal{S}}_u|}, j \in \hat{\mathcal{S}}_u, j \in \mathcal{U}_0\} \mid \Phi_j]}{1 \vee |\hat{\mathcal{R}}_u^{(j)}|}\right]$$

$$\leq \sum_{j\in\mathcal{U}} \mathbb{E}\left[\frac{\alpha|\hat{\mathcal{R}}_u^{(j)}|}{|\hat{\mathcal{S}}_u|} \frac{1}{1 \vee |\hat{\mathcal{R}}_u^{(j)}|}\mathbb{1}\{j \in \hat{\mathcal{S}}_u, j \in \mathcal{U}_0\}\right] = \alpha\mathbb{E}\left[\frac{|\hat{\mathcal{S}}_u \cap \mathcal{U}_0|}{|\hat{\mathcal{S}}_u|}\right]$$

By the construction of the conformal p-value, we analyze each selected unit $j \in \hat{\mathcal{S}}_u$ by conditioning on a carefully constructed quantity $\Phi_j = (\mathcal{D}_{\mathcal{C}\cup\{j\}}^*, \mathcal{D}_{\mathcal{U}\setminus\{j\}})$. It comprises two components: $\mathcal{D}_{\mathcal{U}\setminus\{j\}}$, the test data with the $j$-th sample excluded, and $\mathcal{D}_{\mathcal{C}\cup\{j\}}^* := [Z_i; i \in \mathcal{C} \cup \{j\}]$, the unordered set of labeled data along with the $j$-th sample. The unordered set provides the order statistics but not the specific ordering, which is a common convention in conformal inference literature [35, 34].

Through this approach, we are able to decouple the dependence that arises from the data-dependent selection and the construction of p-values that share the same calibration data. If the selection rule is strongly stable, then $|\hat{\mathcal{S}}_u|$ is fixed given $\Phi_j$ for $j \in \hat{\mathcal{S}}_u$ and the selective conformal p-value in (2) is valid. By performing a careful analysis of the rejection set $\hat{\mathcal{R}}_u$, i.e. replacing it with a pseudo rejection set $\hat{\mathcal{R}}_u^{(j)}$ that remains fixed given $\Phi_j$ and $j \in \hat{\mathcal{R}}_u$, we obtain the finite sample FDR guarantee.

### 3.1 Joint-exchangeable selection

Firstly, we consider the joint-exchangeable selection procedure. The joint-exchangeable selection procedure is applied to $\{X_i : i \in \mathcal{C} \cup \mathcal{U}\}$ with exchangeability, i.e. the selection results remain unchanged after any permutation of data in the merged set $\mathcal{C} \cup \mathcal{U}$, as Definition 3.3 indicates.

**Definition 3.3** (Joint-exchangeable selection). The selection procedure $\mathbf{S}$ is joint-exchangeable with respective to the $\{X_i : i \in \mathcal{C} \cup \mathcal{U}\}$ if

$\mathbf{S}_{\mathcal{D}_c, \mathcal{D}_u}(X_i) = \mathbf{S}_{\mathcal{D}_k, \mathcal{D}_l}(X_i)$ for any $i \in \mathcal{C} \cup \mathcal{U}$ and $\mathcal{D}_k, \mathcal{D}_l$ that are arbitrary partitions of $\mathcal{D}_c \cup \mathcal{D}_u$.

If the selection procedure is independent of both the labeled and test data, it is joint-exchangeable. In the case of the selection with a threshold, the joint-exchangeable selection is equivalent to

$$\tau(X_1, \cdots, X_{|\mathcal{C}\cup\mathcal{U}|}) = \tau(X_{\pi(1)}, \cdots, X_{\pi(|\mathcal{C}\cup\mathcal{U}|)}).$$

where $\tau(\mathcal{D})$ denotes that $\tau$ is computed using the dataset $\mathcal{D}$. Therefore, the joint-exchangeable selection includes selection using constant thresholds or thresholds computed by $\{T_i\}_{i\in\mathcal{C}\cup\mathcal{U}}$ exchangeably. We can verify joint-exchangeable selection is strongly stable through the leaving out condition.

**Proposition 3.4.** *The joint exchangeable selection procedure* $\mathbf{S}_{\mathcal{D}_c, \mathcal{D}_u}$ *is strongly stable.*

According to Theorem 3.2, our procedure ensures FDR control for any joint-exchangeable selection rule, which makes the choice of selection rule quite flexible. For example, we can perform a clustering algorithm on the $\{X_i\}_{i\in\mathcal{C}\cup\mathcal{U}}$ to divide the data into different groups. As a special case, our approach aligns with the InfoSCOP proposed by Gazin et al. [23] under the joint-exchangeable selection rule. They proposed a novel procedure for selecting an informative set and also provided FDR control guarantee as an extension of their FCR control results.

While the joint-exchangeable rule contains various selection strategies, it is worth noting that many cases involve selection that depends solely on the test data. In the following subsections, we investigate several commonly used selection rules that are determined only by the test data. And the assumption in InfoSCOP [23] is not satisfied under these cases.

## 3.2 Top-K/Quantile selection

Next, we consider the top-K or quantile selection rule, which relies solely on the test data $\mathcal{D}_u$. This type of rule is extensively studied in the literature [42, 3, 28] and is commonly used in practice.

Let $\tau_{\text{topK}}$ denote the top-K selection threshold, which is defined as the $(K+1)$-th smallest value in $\{T_j : j \in \mathcal{U}\}$. The top-K rule is equivalent to the quantile selection rule since $\tau_{\text{topK}}$ corresponds to the $(K+1)/m$-quantile of the test data, and the threshold for the $q$-quantile is the $\lceil mq \rceil$-th smallest value. The selected test set and the chosen calibration set under the top-K rule are defined as:

$$\hat{\mathcal{S}}_u = \{j \in \mathcal{U} : T_j < \tau_{\text{topK}}\}, \quad \hat{\mathcal{S}}_c = \{i \in \mathcal{C} : T_i < \tau_{\text{topK}}\}. \tag{3}$$

We can verify that the top-K selection rule is strongly stable by the following proposition. With the support of Theorem 3.2, we can ensure FDR control when employing the top-K selection rule.

**Proposition 3.5.** *For top-K selection rule* **S** *with threshold* $\tau_{\text{topK}}(\mathcal{U})$*, if* $j \in \hat{\mathcal{S}}_u$*, then*

$$\tau_{\text{topK}}(T_{n+1}, \cdots, T_{j-1}, T_j, T_{j+1}, \cdots, T_{n+m}) = \tau_{\text{topK}}(T_{n+1}, \cdots, T_{j-1}, -\infty, T_{j+1}, \cdots, T_{n+m}).$$

*Thus top-K selection rule is strongly stable by the replacing condition.*

## 3.3 General extension to weakly stable selection

In this subsection, we consider weakening the strongly stable condition. For example, the mean thresholding rule does not satisfy the strong stability. By the insight of our proof, the key requirement is the property of $\hat{\mathcal{S}}_u$ such that we can handle the randomness of the selection event. Hence we define the weakly stable selection rule as follows:

**Definition 3.6** (Weak stability). Given selection set $\hat{\mathcal{S}}_u = \{j \in \mathcal{U} : \mathbf{S}_{\mathcal{D}_c,\mathcal{D}_u}(X_j) = 1\}$. We call the selection rule $\mathbf{S}_{\mathcal{D}_c,\mathcal{D}_u}$ is weakly stable if

$$\mathbf{S}_{\mathcal{D}_c,\mathcal{D}_u}(X_i) = \mathbf{S}_{\mathcal{D}_c \cup \{Z_j\}, \mathcal{D}_u \setminus \{Z_j\}}(X_i) \text{ for any } j \in \hat{\mathcal{S}}_u \text{ and any } i \in \mathcal{U}.$$

The weak stability does not require the $\mathbf{S}_{\mathcal{D}_c,\mathcal{D}_u}(X_i) = \mathbf{S}_{\mathcal{D}_c \cup \{Z_j\}, \mathcal{D}_u \setminus \{Z_j\}}(X_i)$ hold for $i \in \mathcal{C}$. Except for the selection rules previously discussed, the commonly used mean selection rule by test data only is also weakly stable, i.e. $\{j \in \mathcal{U} : T_j < \frac{1}{m} \sum_{i \in \mathcal{U}} T_i\} = \{j \in \mathcal{U} : T_j < \frac{1}{m-1} \sum_{i \in \mathcal{U} \setminus \{j\}} T_i\}$.

As the weakly stable rule fails to guarantee the exchangeability of the selected calibration set and test set, it motivates us to explore a new construction method for selective p-values. The selection set is denoted as $\hat{\mathcal{S}}_u = \{j \in \mathcal{U} : \mathbf{S}_{\mathcal{D}_c,\mathcal{D}_u}(X_j) = 1\}$. By the definition of weakly stable selection, we know that $\hat{\mathcal{S}}_u = \{j \in \mathcal{U} : \mathbf{S}_{\mathcal{D}_c \cup \{Z_j\}, \mathcal{D}_u \setminus \{Z_j\}}(X_i) = 1\}$ is also true. Specially, if the selection rule is determined only by the test data, it holds that $\mathbf{S}_{\mathcal{D}_c \cup \{Z_j\}, \mathcal{D}_u \setminus \{Z_j\}}(\cdot) = \mathbf{S}_{\mathcal{D}_u \setminus \{Z_j\}}(\cdot)$.

Therefore, we adaptively pick data from the labeled set using the same "leaving out" rule as selecting $Z_j$. For any $i \in \mathcal{C}$ and $j \in \mathcal{U}$, $\{\mathbf{S}_{\mathcal{D}_u \setminus \{Z_j\}}(X_i) = 1\}$ and $\{\mathbf{S}_{\mathcal{D}_u \setminus \{Z_j\}}(X_j) = 1\}$ are symmetric to $X_i$ and $X_j$. Leveraging this, we can pick up the calibration set by

$$\hat{\mathcal{S}}_c(j) = \{i \in \mathcal{C} : \mathbf{S}_{\mathcal{D}_u \setminus \{Z_j\}}(X_j) = 1\} \text{ for } j \in \hat{\mathcal{S}}_u$$

and construct the adaptive selective conformal p-value as

$$p_j^{\text{adapt}} := \frac{1 + |\{i \in \hat{\mathcal{S}}_c(j) \cap \mathcal{C}_0 : V_i \leq V_j\}|}{1 + |\hat{\mathcal{S}}_c(j) \cap \mathcal{C}_0|}. \tag{4}$$

For example, the mean selection rule picks up calibration set by $\hat{\mathcal{S}}_c(j) = \{i \in \mathcal{C} : T_i \leq \frac{1}{m-1} \sum_{k \in \mathcal{U} \setminus \{j\}} T_k\}$. Our adaptive strategy shares the same goal as the swapping strategy [? 28, 4] in terms of constructing valid p-value after selection. However, our approach is different from the others in core motivations since ours is directly related to weak stability, leading to a faster computation and a more intuitive explanation here. We can verify that $p_j^{\text{adapt}}$ is valid since $\{Z_k\}_{k \in \hat{\mathcal{S}}_c(j) \cup \{j\}}$ are exchangeable.

**Proposition 3.7.** *The adaptive selective conformal p-value* $p_j^{\text{adapt}}$ *for weakly stable selection which is determined only by the test data satisfies* $\Pr(p_j^{\text{adapt}} \leq t \mid j \in \hat{\mathcal{S}}_u, j \in \mathcal{U}_0) \leq t$.

However, the p-value for each selected test point $p_j^{\text{adapt}}$ is based on a different calibration set $\hat{\mathcal{S}}_c(j)$, making the dependence structure intricate. Consequently, the BH procedure has no safe guarantee to control the FDR. But we find that BH is robust and can produce satisfactory results empirically.

Although this observation is acceptable, it would be desirable to design a new procedure to guarantee the finite sample FDR control. To remedy this, we employ the conditional calibration framework [22] to achieve finite sample FDR control. The overall procedure can be re-framed as the e-BH framework [58] and a recent novel approach for boosting the power of e-BH procedure [33] can be employed in our setting. The details are displayed in Appendix B.

Under the weakly stable selection rule, our method differs fundamentally from InfoSCOP [23] in both methodology and theory. Since our approach and InfoSCOP are designed for different goals, resulting in different analytical frameworks. Ours is specifically designed to address the multiple testing problem across various selection rules. From the perspective of conditional calibration, our method is unified, where the BH procedure for strongly stable selection can be seen as a special case. As a comparison, InfoSCOP stands out as a remarkable work for selecting an informative set with FCR control, but it is not primarily designed for our problem, which limits their method's applicability to more general selection rules.

## 4 Numerical studies

To demonstrate the wide applicability of the proposed method in Algorithm 1, we conduct comprehensive numerical studies. For regression setting, the region for the hypothesis is $\mathcal{A} = \{y : y > c_0\}$, where $c_0$ is a fixed constant. For classification, we set $\mathcal{A} = \{1\}$, i.e. the class 1 as the target region, and we denote the prediction $\hat{\mu}(X)$ as the predicted probability of $Y = 1$. The nonconformity score we use to construct conformal p-value for both settings is $V(\hat{\mu}(X_j)) = -\hat{\mu}(X_j)$.

**Benchmarks:**  Since selective multiple testing has not been investigated before, we consider several intuitive methods as comparing benchmarks.

- SCPV: Our procedure in Algorithm 1. Specially, for mean selection rule, we use the adaptive p-value in (4) along with the BH procedure. The results for using conditional calibration can be found in Appendix B;

- OMT: Ordinary multiple testing which constructs the conformal p-value directly as in equation (1) for each selected sample based on the entire labeled set.

- AMT (BH/BY): An intuitive procedure by multiplying (1) with the selection proportion of null samples. As the adjusted p-values have intractable dependence, making the validity of BH procedure suspicious, we also utilize the Benjamini-Yekutieli (BY) [11] procedure to control the FDR. More details are provided in Appendix A.1.

- SCOP: Directly invert the selective prediction interval constructed by Bao et al. [3] into a test and make decision by whether the $c_0$ is contained in the interval. It is designed for regression setting and does not have FDR guarantee. See more detail in Appendix A.2.

We also use the Storey's method [49] to increase power. See more information in Appendix C.1.

**Selection rule:**  In the numerical studies, we choose the selection statistic $T_i$ based on a specific component of $X$. The selected subset is $\hat{\mathcal{S}}_u = \{i \in \mathcal{U} : T_i < \hat{\tau}\}$, where $\hat{\tau}$ is the threshold. Three different choices of selection thresholds are considered.

- **Exchangeable (Exch):** 70%-quantile of the first component of $X$ in both labeled set and test set, that is $\hat{\tau}$ is the 70%-quantile of $\{T_i : i \in \mathcal{C} \cup \mathcal{U}\}$.

- **Quantile (Quan):** 70%-quantile of the first component of $X$ in the test set, that is $\hat{\tau}$ is the 70%-quantile of $\{T_i : i \in \mathcal{U}\}$.

- **Mean:** the sample mean of the first component of $X$ in the test set, that is $\hat{\tau} = \frac{1}{m} \sum_{j \in \mathcal{U}} T_j$.

**Evaluation metrics:**  We empirically evaluate the FDR by averaging the FDP based on selected samples and the power by averaging the proportion of correct selections among all selected alternative test samples, i.e. Power $:= |i \in \hat{\mathcal{S}}_u : i \in \mathcal{R}, Y_i > c_0|/|i \in \hat{\mathcal{S}}_u : Y_i > c_0|$ over 100 independent runs.

## 4.1 Results on synthetic data

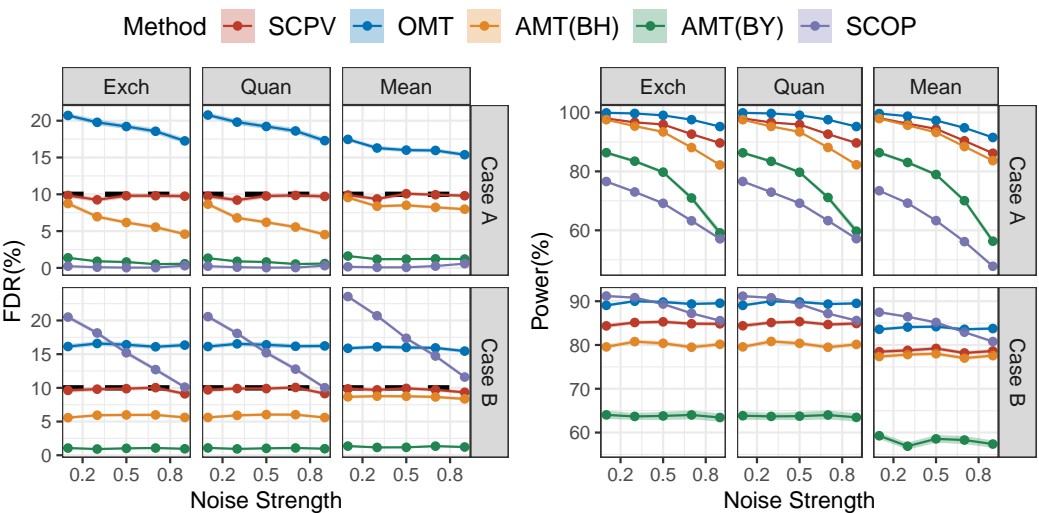

Figure 1: Empirical FDR (left) and Power (right) of five methods under different scenarios and selection rules. The Noise Strength varies from $0.1$ to $1$. The black dashed line in the left plot denotes the target FDR level $\alpha = 10\%$.

In synthetic studies, we generate i.i.d. 10-dimensional covariates from $X_i \sim \text{Unif}([-1, 1])^{10}$. The corresponding regression responses are generated as $Y_i = \mu(X_i) + \epsilon_i$, where $\epsilon_i$ denotes independent random noise. The following data-generating scenarios are considered:

- **Case A:** The data generating model is $\mu(X) = 4(X^{(1)} + 1)|X^{(1)}|\mathbb{1}\{X^{(2)} > -0.4\} + 4(X^{(1)} - 1)\mathbb{1}\{X^{(2)} \leq -0.4\}$. The noise is $\epsilon_i \sim N(0, \sigma^2)$, independent of $X$. And $c_0 = 2$.
- **Case B:** $\mu(X) = \mathbb{1}\{X^T\beta > 1.5\}$, where $\beta = (1, -1, 2, -2, 0, 0, 0, 0, 0, 0)$. The noise is $\epsilon_i \sim N(0, 0.1\sigma^2)$, independent of $X$. And $c_0 = 0.12$.

We fix the labeled data size $n = 1,200$ and the unlabeled data size $m = 1,200$. We fit the regression models $\hat{\mu}(\cdot)$ on an additional labeled set with size $1,200$ using the random forest algorithm, implemented by R package `randomForest` with default parameters. Specifically, for both scenarios, we select the first component of $X$ as the selection statistic, i.e., $T_i = X_i^{(1)}$.

Figure 1 displays the FDR (left) and power (right) through varying noise strength. Across both settings, SCPV can deliver valid FDR control. As expected, the OMT fails to control FDR. This can be understood since the OMT constructs conformal p-values without consideration of the selection procedure, leading to smaller p-values possibly. Moreover, our method demonstrates greater statistical power compared to AMT. This is because AMT does not make full use of information from the selection procedure. Meanwhile, SCOP fails to control FDR in case B. And even if SCOP can control FDR in case A, the accompanying loss of power is substantial. This is because the SCOP is not designed for multiple testing and can not deliver valid FDR results.

## 4.2 Results on real data

We consider several real data experiments including both regression (Reg) and classification (Cla) settings. We summarize the datasets in Table 1. The test samples and labeled samples are constructed by subsampling the dataset with $n = 1000$ and $m = 2000$, and the null proportion is fixed by $\pi = 0.8$. We sam-

Table 1: Summary of real-world datasets for conformalized multiple testing

|  | Abalone[37] | Census[9] | Credit[30] | Promotion[36] |
|---|---|---|---|---|
| #Features | 8 | 14 | 30 | 12 |
| #Instances | 4,177 | 48,842 | 284,808 | 54,809 |
| Task | Reg | Cla | Cla | Cla |

ple another 1000 samples to train a random forest model for classification and regression. See more details in Appendix C.2. The results are reported in Table 2. The AMT(BY) outputs null rejection set in most cases, hence we omit it. As expected, SCPV achieves highest power among methods controlling the FDR, verifying its effectiveness and validity.

Table 2: Empirical FDR (%) and Power (%) with target FDR $\alpha = 10\%$. The bracket contains the standard error (%). The highest power among methods controlling the FDR is bolded.

| DATASET | METHOD | EXCH | | QUAN | | MEAN | |
|---|---|---|---|---|---|---|---|
| | | FDR | POWER | FDR | POWER | FDR | POWER |
| ABALONE | SCPV | 6.68(0.75) | **11.3(0.13)** | 6.61(0.75) | **11.2(1.3)** | 6.37(0.73) | **7.92(0.96)** |
| | OMT | 24.8(0.51) | 51.7(0.86) | 24.9(0.51) | 51.8(0.88) | 20.1(0.65) | 30.9(1.2) |
| | AMT(BH) | 5.12(0.65) | 8.30(1.0) | 5.13(0.65) | 8.30(1.1) | 6.09(0.71) | 7.50(0.90) |
| CENSUS | SCPV | 7.12(0.80) | **15.3(1.5)** | 7.04(0.75) | **15.2(1.5)** | 7.20(0.69) | **15.7(1.3)** |
| | OMT | 13.9(0.68) | 30.3(1.2) | 14.0(0.68) | 30.4(1.2) | 14.7(0.57) | 32.6(1.1) |
| | AMT(BH) | 2.55(0.49) | 5.48(0.86) | 2.48(0.49) | 5.26(0.85) | 6.63(0.65) | 12.4(1.1) |
| CREDIT | SCPV | 8.85(0.54) | **85.9(0.33)** | 8.85(0.54) | **85.9(0.33)** | 9.15(0.72) | **84.5(0.35)** |
| | OMT | 12.6(0.62) | 86.7(0.33) | 12.7(0.61) | 86.7(0.33) | 14.9(0.70) | 85.9(0.36) |
| | AMT(BH) | 3.46(0.31) | 84.9(0.32) | 3.46(0.31) | 84.9(0.32) | 2.77(0.28) | 83.0(0.38) |
| PROMOTION | SCPV | 7.44(0.61) | **19.6(1.1)** | 7.67(0.62) | **19.7(1.1)** | 7.55(0.64) | **14.3(0.90)** |
| | OMT | 19.7(0.68) | 35.7(0.65) | 19.7(0.69) | 35.6(0.66) | 18.3(0.68) | 25.4(0.58) |
| | AMT(BH) | 5.33(0.52) | 16.5(1.0) | 5.45(0.52) | 16.8(1.0) | 6.07(0.57) | 13.1(0.85) |

## 5 Limitations and discussions

Here we point out the current limitations of our paper and discuss the potential directions. First, our work relies on the i.i.d. assumption. Exploring the selective multiple testing problem in scenarios where the labeled set and test set exhibit different distributions would be interesting. Second, we require the selection rule to be stable for theoretical guarantee. It would be attractive to consider complex selection procedures that lack stability, such as clustering based on test data only.

## Acknowledgments and Disclosure of Funding

We thank anonymous area chair and reviewers for their helpful comments. Zou was supported by the National Key R&D Program of China (Grant Nos. 2022YFA1003703, 2022YFA1003800), the National Natural Science Foundation of China (Grant Nos. 11925106, 12231011, 11931001, 12226007, 12326325).

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

# A Details of the comparing benchmarks

## A.1 The adjusted p-value (AMT)

Here, we provide an overview of the statistical properties associated with the adjusted p-values.

We draw inspiration from the work of Benjamini and Yekutieli [12], which constructs adjusted confidence intervals for selected parameters for FCR control. They multiplies the confidence level $\alpha$ by a quantity related to the proportion of selected candidates over all candidates. In most cases, the quantity is approximately the selection proportion. And they proved such simple adjustment on the level can provide FCR control.

Analogously, we can adjust the p-value after selection by multiplying the selection proportion:

$$p_j^* = \min\left\{\hat{\theta}\frac{1 + |\{i \in \mathcal{C}_0 : V_i \leq V_j\}|}{1 + |\mathcal{C}_0|}, 1\right\}, \quad j \in \hat{\mathcal{S}}_u. \tag{5}$$

where $\hat{\theta}$ represents an estimator that estimates the selected proportion under the null hypotheses. Since the response of the test data is not directly observable, we estimate this proportion by employing the same selection procedure on the labeled data, i.e. $\hat{\theta} = \frac{|\mathcal{C}_0|}{|\mathcal{C}_0 \cap \hat{\mathcal{S}}_c|}$. We can verify that the adjusted p-value is super-uniform for joint-exchangeable selection rule.

**Proposition A.1.** *The adjusted p-value is super-uniform i.e.*

$$\mathbb{P}(p_j^* \leq \alpha \mid j \in \mathcal{U}_0 \cap \hat{\mathcal{S}}_u) \leq \alpha$$

where $p_j^*$ is defined in (8).

*Proof.* The case $p^* = 1$ is trivial, so we only consider the case $\hat{\pi}\frac{1 + |\{i \in \mathcal{C}_0 : V_i \leq V_j\}|}{1 + |\mathcal{C}_0|} \leq 1$. In this case, we have

$$\mathbb{P}\left(p_j^* \leq \alpha \mid j \in \mathcal{U}_0 \cap \hat{\mathcal{S}}_u\right) = \mathbb{P}\left(\frac{|\mathcal{C}_0|}{|\mathcal{C}_0 \cap \hat{\mathcal{S}}_c|}\frac{1 + |\{i \in \mathcal{C}_0 : V_i \leq V_j\}|}{1 + |\mathcal{C}_0|} \leq \alpha \mid j \in \mathcal{U}_0 \cap \hat{\mathcal{S}}_u\right)$$

$$\leq \mathbb{P}\left(\frac{1 + |\{i \in \mathcal{C}_0 : V_i \leq V_j\}|}{1 + |\mathcal{C}_0 \cap \hat{\mathcal{S}}_c|} \leq \alpha \mid j \in \mathcal{U}_0 \cap \hat{\mathcal{S}}_u\right)$$

$$\leq \mathbb{P}\left(\frac{1 + |\{i \in \mathcal{C}_0 \cap \hat{\mathcal{S}}_c : V_i \leq V_j\}|}{1 + |\mathcal{C}_0 \cap \hat{\mathcal{S}}_c|} \leq \alpha \mid j \in \mathcal{U}_0 \cap \hat{\mathcal{S}}_u\right)$$

$$\overset{(i)}{\leq} \alpha$$

where (i) use the property that the variables $\{V_j : j \in \mathcal{U}_0 \cap \hat{\mathcal{S}}_u\}$ and $\{V_i : i \in \mathcal{C}_0 \cap \hat{\mathcal{S}}_c\}$ are exchangeable when the selection procedure is exchangeable of the test and labeled data. $\square$

The correlation of the adjusted p-values can be complex when using an arbitrary selection procedure. In order to address this issue, we propose utilizing the Benjamini-Yekutieli (BY) [11] procedure to effectively control the FDR. It replaces the original level $\alpha$ with $\alpha/L_{|\hat{\mathcal{S}}_u|}$, where

$$L_{|\hat{\mathcal{S}}_u|} = \sum_{i=1}^{|\hat{\mathcal{S}}_u|} \frac{1}{i} = \log|\hat{\mathcal{S}}_u| + O(1).$$

The BY method can handle the dependence between p-values, but deliver a more conservative result.

## A.2 Selection conditional conformal prediction (SCOP)

The selection conditional conformal prediction (SCOP) proposed by Bao et al. [3] is a method for constructing valid prediction intervals after selection. The prediction interval is reported only when it is selected. The SCOP aims to control the false coverage-statement rate (FCR) [12], which is the

expected ratio of the number of selected prediction intervals failing to cover their respective true outcomes to the total number of selected prediction interval, i.e.

$$\text{FCR} := \mathbb{E}\left[\frac{\sum_{j \in \mathcal{U}} \mathbb{I}\{j \in \hat{\mathcal{S}}_u, Y_j \notin \text{PI}(X_j)\}}{1 \vee |\hat{\mathcal{S}}_u|}\right],$$

where $\text{PI}(X_j)$ is the prediction interval. To ensure FCR control, the SCOP involves a similar procedure to pick up a calibration set from the labeled data using the same selection rule. And then the prediction interval for selected individual is constructed via the residuals in picked calibration set.

As a natural idea, we can simply invert the prediction interval into a hypothesis testing. If the hypothesis is $H_{0,j} : Y_j \leq c_0$ v.s. $H_{0,j} : Y_j > c_0$, reject the single hypothesis with type I error at $\alpha$ is equivalent to that the one sided prediction interval covers $c_0$. We formulate this idea as an intuitive benchmark as Algorithm 2.

---

**Algorithm 2** SCOP for selective multiple testing

---

**Input:** Labeled set $\mathcal{D}_c$, test set $\mathcal{D}_u$, selection procedure $\mathbf{S}_{\mathcal{D}_c,\mathcal{D}_u}$, prediction model $\hat{\mu}(\cdot)$, FCR level $\alpha \in (0, 1)$.

    **Step 1** Apply the selective procedure $\mathbf{S}$ to obtain the selected subsets $\hat{\mathcal{S}}_u$ and $\hat{\mathcal{S}}_c$.

    **Step 2** Compute residuals $\{R_i = Y_i - \hat{\mu}(X_i) : i \in \hat{\mathcal{S}}_c\}$.

    **Step 3** Construct selective conformal prediction intervals for each $j \in \hat{\mathcal{S}}_u$ by

$$\text{PI}(X_j) = (-\infty, \hat{\mu}(X_j) + Q_\alpha(\{R_i\}_{i \in \hat{\mathcal{S}}_c})],$$

    where $Q_\alpha(\{R_i\}_{i \in \hat{\mathcal{S}}_c})$ denotes the $\lceil(1-\alpha)(|\hat{\mathcal{S}}_c|+1)\rceil$-th smallest value in $\{R_i\}_{i \in \hat{\mathcal{S}}_c}$.

    **Step 4** Reject sample $j$ if $c_0 \in \text{PI}(X_j)$

**Output:** Rejection set $\hat{\mathcal{R}}^{\text{SCOP}} = \{j \in \hat{\mathcal{S}}_u : c_0 \in \text{PI}(X_j)\}$.

---

Under the null, we have $\mathbb{P}(c_0 \notin \text{PI}(X_j)) \leq \mathbb{P}(Y_j \notin \text{PI}(X_j)) \leq \alpha/\mathbb{P}(Y_j \leq c_0)$. This implies marginal coverage, as for a single hypothesis $H_{0j}$, we can control the type I error. It is important to note that the scaling of the inequality $\mathbb{P}(c_0 \notin \text{PI}(X_j)) \leq \mathbb{P}(Y_j \notin \text{PI}(X_j))$ is often too conservative.

However, when it comes to simultaneous testing, the SCOP procedure fails to control the FDR. This is because the FDR is built up on the rejection set $\hat{\mathcal{R}} \subset \hat{\mathcal{S}}_u$, where the rejection decisions are intricately linked to the entire selection set and the inherent randomness within the selection set further complicates the distribution of the final rejection set. Hence our work address this challenge by carefully analyzing the randomness from selection and rejection set.

## B  Conditional calibration

Here we introduce the frame work of conditional calibration [22].

The first idea of conditional calibration is to control a conditional expectation, given some conditioning statistic $\Phi_j$ that blocks most or all of the nuisance parameters from influencing the conditional analysis. And we require that $p_j^{\text{adapt}}$ is conditionally superuniform given $\Phi_j$:

$$\Pr(p_j^{\text{adapt}} \leq t \mid j \in \hat{\mathcal{S}}_u, j \in \mathcal{U}_0, \Phi_j) \leq t$$

Secondly, the number of rejections should should be bound from below by a known function of $\Phi_j$. If all constraints are satisfied, that set of rejections is guaranteed to control the FDR bellow $\alpha$.

**Step 1: Calibration** For each of the $m$ test points, $i \in \mathcal{D} = \{n+1, ..., n+m\}$, let $\Phi_j = (\mathcal{D}^*_{\mathcal{C} \cup \{j\}}, \mathcal{D}_{\mathcal{U} \setminus \{j\}})$. For each test point $j \in \hat{\mathcal{S}}_u$, compute the adaptive p-value

$$p_\ell^{\text{adapt},(j)} = \frac{\sum_{i \in \hat{\mathcal{S}}_c \cap \mathcal{C}_0} \mathbb{1}\{V_i < V_\ell\} + \mathbb{1}\{V_j < V_\ell\}}{1 + |\hat{\mathcal{S}}_c \cap \mathcal{C}_0|}, \quad \forall \ell \neq j, \quad \ell \in \hat{\mathcal{S}}_u. \tag{6}$$

Next, let $\hat{R}_i$ indicate the number of rejections obtained by applying BH at level $\alpha$, for some fixed $\alpha \in (0, 1)$, to the approximate p-values $\{p_l^{\text{adapt},(j)} : \ell \neq j, \ell \in \hat{\mathcal{S}}_u\} \cup \{0\}$.

**Step 2: Preliminary rejection.** Define the preliminary rejection set $\mathcal{R}_+$ as:

$$\mathcal{R}_+ = \left\{ i \in \hat{\mathcal{S}}_u : p_i^{\mathrm{adapt}} \leq \frac{\alpha \hat{R}_i}{|\hat{\mathcal{S}}_u|} \right\}$$

$\hat{R}_+ = |\mathcal{R}_+|$. If $\hat{R}_+ \geq \hat{R}_i$ for all $i \in \mathcal{R}_+$, then return the final rejection set $\mathcal{R} = \mathcal{R}_+$. Otherwise, proceed to the next step.

**Step 3: Pruning.** (a) Deterministic pruning: Define $R$ as:

$$R_{dtm} = \max \left\{ r : \left| i \in \mathcal{R}_+ : \hat{R}_i \leq r \right| \geq r \right\}.$$

The pruned rejection set $\mathcal{R}$ is that containing the indices with $i \in \mathcal{R}_+$ and $\hat{R}_i < R$.

(b) Randomized pruning: Generate independent standard uniform random variables $\epsilon_i$ for each $i \in \mathcal{R}_+$, and define $R$ as:

$$R_{rdm} = \max \left\{ r : \left| i \in \mathcal{R}_+ : \epsilon_i \leq r/\hat{R}_i \right| \geq r \right\}.$$

The pruned rejection set $\mathcal{R}$ is the set containing the indices $i \in \mathcal{R}_+$ such that $\epsilon_i < R/\hat{R}_i$.

The conventional conditional calibration offers a flexible framework to decouple the dependence between p-values. But in our selective setting, the number of test units is $|\hat{\mathcal{S}}_u|$, which can be complicatedly dependent with both $p_j$ and $\hat{\mathcal{R}}_j$. And when analyzing the FDR, the event that j-th sample is selected is also involved. So our primary focus is on ensuring $\mathbb{E}\left[ \frac{\mathbb{1}\{p_j \leq \frac{\alpha \hat{R}_i}{|\hat{\mathcal{S}}_u|}, j \in \hat{\mathcal{S}}_u\}|\hat{\mathcal{S}}_u|}{|\hat{\mathcal{R}}_j|} \mid \Phi_j \right] \leq \alpha$.

The conditional calibration framework primarily focuses on the correlation of p-values. However, a significant challenge arises because FDR control in a selective setting involves not only individual p-values but also the selection procedure itself. Consequently, the selective effects are unavoidable when implementing conditional calibration. To address this, we leverage the stability property of the selection rule, which allows us to effectively conduct analysis over the selected subset effectively and rigorously.

We can prove that the conditional calibration applied to the adaptive selective conformal p-values can control the FDR at $\alpha$. The technical proofs are deferred in Appendix F.8.

**Theorem B.1.** *Assume the data are i.i.d. and the selection rule is weakly stable. Then, the FDR output by the above three-step procedure applied to $p^{\mathrm{adapt}}$ is smaller than $\alpha \mathbb{E}\left[ \frac{|\hat{\mathcal{S}}_u \cap \mathcal{U}|_0}{|\hat{\mathcal{S}}_u|} \right]$.*

It is easy to observe that $R_{dtm} \subseteq R_{rdm}$, indicating that randomized pruning results in larger rejection sets. Therefore, we employ randomized pruning in practice to enhance power. By our empirical investigations, we find the BH procedure applied to $p_j^{\mathrm{adapt}}$ can ideally control the FDR, and the conditional calibration approach with random pruning also has a close performance in power. Figure 2 displays the FDR (left) and power (right) through varying noise strength employing conditional calibration and BH procedure.

## B.1 Eliminating randomness by boosting e-BH

The deterministic pruning process would lose certain power. Although randomized pruning can improve this situation, the external randomness can potentially hinder the reproducibility of the results (the procedure can be quite sensitive to the realization of the $\epsilon$'s, leading to different selections across various algorithm runs). A recent method [33] can enhance the power of the pruning process without introducing additional randomness.

Specifically, let $\mathcal{R}(\boldsymbol{e})$ represent the rejection set yielded by the e-BH procedure on $\boldsymbol{e}$ at level $\alpha \in (0, 1)$. For each $j \in [m]$, define $\hat{\mathcal{R}}_j(\boldsymbol{e}) := \mathcal{R}(\boldsymbol{e}) \cup \{j\}$ and subsequently define the function

$$\phi_j(c; S_j) := \mathbb{E}\left[ \frac{m}{\alpha} \cdot \frac{\mathbb{1}\left\{ c\tilde{e}_j \geq \frac{m}{\alpha|\hat{\mathcal{R}}_j(\tilde{e})|} \right\}}{\left| \hat{\mathcal{R}}_j(\tilde{e}) \right|} - \tilde{e}_j \;\middle|\; S_j \right]$$

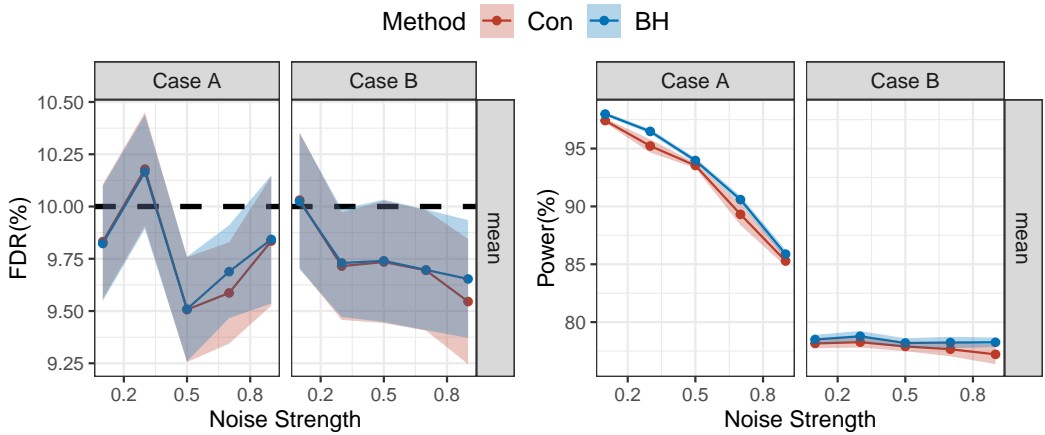

Figure 2: Empirical FDR (left) and Power (right) of conditional calibration (Con) and BH procedure under different cases for the mean selection rule. The black dashed line denotes the target FDR level 10%.

where $\widetilde{e} = (\widetilde{e}_1, \ldots, \widetilde{e}_m)$ follows the conditional distribution $e \mid S_j$. With the associated critical value $\widehat{c}_j := \sup\{c : \phi_j(c; S_j) \leq 0\}$, the boosted e-values are constructed as:

$$
e_j^{\flat} = \begin{cases}
\frac{m}{\alpha|\widehat{\mathcal{R}}_j(e)|} \cdot \mathbb{1}\left\{\widehat{c}_j e_j \geq \frac{m}{\alpha|\widehat{\mathcal{R}}_j(e)|}\right\} & \text{if } \phi_j(\widehat{c}_j; S_j) \leq 0 \\
\frac{m}{\alpha|\widehat{\mathcal{R}}_j(e)|} \cdot \mathbb{1}\left\{\widehat{c}_j e_j > \frac{m}{\alpha|\widehat{\mathcal{R}}_j(e)|}\right\} & \text{if } \phi_j(\widehat{c}_j; S_j) > 0
\end{cases}
$$

They prove that the boosted e-values are generalized e-values and $\mathcal{R}(e) \subseteq \mathcal{R}(e^b)$.

Our procedure can be viewed as a generalization for a selective scenario of their approach. The conditional calibration approach with deterministic pruning is equivalent to the e-BH procedure applied to $\{e_j : j \in \hat{\mathcal{S}}_u\}$, where $e_j = \frac{|\hat{\mathcal{S}}_u| \mathbb{1}\{p_j \leq \frac{\alpha \hat{R}_j(\mathbf{p})}{|\hat{\mathcal{S}}_u|}\}}{\alpha \hat{R}_j(\mathbf{p})}$ by referencing Jin and Candès [27] and $\mathbf{p} = \{p_j\}_{j \in \hat{\mathcal{S}}_u}$. Another form of e-value existing in conformal inference is based on a specific stopping time [43, 7]. But we find it is not directly applicable in our setting as our calibration sets are different for each test data. Under our stability assumption, we can confirm that $e_j$ is a valid e-value in a manner similar to Lemma E.2 in our paper. With this equivalence property, the boosting method can be directly applied to our deterministic pruning approach by constructing the new boosted e-value with $m = |\hat{\mathcal{S}}_u|$ and $S_j = (\mathcal{D}_{\mathcal{C} \cup \{j\}}^*, \mathcal{D}_{\mathcal{U} \setminus \{j\}})$.

## C   Details for the numerical experiments

### C.1   Implementation of Storey's method

The Storey's method [49] aims to estimate the null proportion $\pi$ to increase the detection power. In our setting, the null proportion can be directly estimated by the corresponding proportion in the labeled set, i.e. $\hat{\pi}^{\mathrm{OMT}} = \hat{\pi}^{\mathrm{AMT}} = |\mathcal{C}_0|/|\mathcal{C}|$ and $\hat{\pi}^{\mathrm{SCPV}} = |\hat{\mathcal{S}}_c \cap \mathcal{C}_0|/|\hat{\mathcal{S}}_c|$. And when applying BH procedure, we will use a level of $\alpha/\hat{\pi}$ instead of $\alpha$, such that the FDR can be controlled at $\alpha$ exactly.

### C.2   Details of the real data experiments

- **Abalone** [37]: contains easily obtainable measurements of abalone. The task is to predict the age of abalone from physical measurements. We use the shell weight as the selection score. The $c_0$ we used for this task is taken by 12.
- **Census** [9]: contains census data extracted from 1994 Census Bureau database. We focus on people from America and regard the income attribute as the response of interest, which is

a binary variable indicating whether one's income exceeds \$50K per year. The feature of age is used as selection score $T$.

- **Credit** [30]: contains transactions made by credit cards over the course of two days, some being frauds. The task is to identify the frauds and we use the specific feature, amount, as the selection score. Since it contains only 492 samples of class 1, we set the null proportion at 0.9 instead.

- **Promotion** [36]: contains employee's past and current performance and the final promotions. The task is to predict whether a potential promotee at checkpoint in the test set will be promoted or not after the evaluation process. We use the specific feature, average score in current training evaluations, as the selection score.

# D  Additional comparing methods

We discuss two additional comparing methods which are nicely suggested by the reviewers.

**Self-consistent/compliant adjustment (SCA)**   Using the marginal p-values in 1, one can directly achieve FDR control under any data-dependent selection simply by taking the largest self-consistent rejection set, i.e. the largest subset $\mathcal{R}$ s.t. $p_i \leq \alpha' \mathcal{R}/K$ for each $i \in \mathcal{R} \subseteq \hat{\mathcal{S}}_u$, where $\alpha'$ is the largest value that satisfies $\pi_0 \alpha' \left(1 + \log\left(1/\left(\pi_0 \alpha'\right)\right)\right) \leq \alpha$ where $\pi_0 = |\mathcal{C}_0|/|\mathcal{C}|$ is the null proportion. This is a direct consequence of Theorem 3 of [51] and the PRDS property of conformal p-values from [8].

We analyze the comparison between our approach and the baseline method from two perspectives. From the theoretical point of view, we have observed that the power loss associated with utilizing a selective conformal p-value is usually less than that incurred by the FDR-Linking method. To illustrate this, assume $\pi_0 = 0.7$ and $\alpha = 0.1$ as in the simulation setting of quantile selection, then we derive $\alpha' \approx 0.025$. The AMT method adjusts the marginal p-value after selection by multiplying the selection proportion $\hat{\theta} = 1/0.7$. This is equivalent to employing the BH procedure on the marginal p-value with $\alpha = 0.07$, which evidently yields greater power than SCA. Additionally, AMT does not make full use of the information from the selection procedure. In contrast, our proposed method uses a smaller p-value than AMT, which suggests more power increase.

In terms of empirical performance, as demonstrated in both cases from our paper, the SCA method suffers from a power loss, confirming our theoretical analysis.

Table 3: Comparisons of empirical FDR (%) and Power (%) with target FDR level $\alpha = 10\%$ by 500 repetitions.

|  |  | QUAN | | MEAN | |
| --- | --- | --- | --- | --- | --- |
|  |  | FDR | POWER | FDR | POWER |
| CASE A | SCA | 3.59 | 88.7 | 2.85 | 84.7 |
|  | SCPV | 9.83 | 93.9 | 9.90 | 93.9 |
|  | AMT | 6.23 | 92.1 | 8.28 | 92.6 |
| CASE B | SCA | 3.79 | 75.7 | 3.38 | 66.4 |
|  | SCPV | 9.82 | 84.9 | 9.79 | 81.1 |
|  | AMT | 8.71 | 77.0 | 5.81 | 79.5 |

**InfoSCOP**   Gazin et al. [23] propose a novel method named InfoSCOP, which is closely related to our approach under the joint-exchangeable selection rule. Below, we provide a detailed discussion of their method. The primary objective of InfoSCOP is to select informative prediction sets with false coverage rate (FCR) control, although it is not specifically designed for multiple testing.

Their focus is on an informative selective prediction set procedure, denoted as $\hat{\mathcal{S}}_u^{\text{info}} \subset \mathcal{U}$, where each prediction set $C_j$ is $\mathcal{I}$-informative for every $j \in \hat{\mathcal{S}}_u^{\text{info}} \subset \mathcal{U}$. If a prediction set $C_j^\alpha$ is $\mathcal{I}$-informative, then all the prediction sets it contains are also $\mathcal{I}$-informative, and it is right-continuous for the

coverage level. By leveraging the property of $\mathcal{I}$-informative selection, they link the FCR control problem with BH procedure, and verify that the FCR control can also imply the following FDR control:

$$\mathbb{E}\left[\frac{\sum_{j\in\hat{\mathcal{S}}_u^{\text{info}}} \mathbb{1}\{Y_j \notin \cup_{C\in\mathcal{I}} C\}}{1 \vee \hat{\mathcal{S}}_u^{\text{info}}}\right].$$

InfoSCOP's procedure to achieve FCR control consists of two main parts. The first part transforms the informative selection procedure into a specific BH procedure. Let $\mathbf{p} = \{p_j\}_{j\in\mathcal{U}}$ denote the set of p-values over test set. These p-values are inverted into $\mathcal{I}$-adjusted p-values by

$$q_j = \min\{\alpha \in (0,1] : C_j^\alpha(\mathbf{p})\}.$$

The BH procedure is then applied to the $\mathbf{q} = \{q_j\}_{j\in\mathcal{U}}$ to obtain a selection set $\text{BH}(\mathbf{q})$. The second part involves using the adjusted approach from Benjamini and Yekutieli [12] to construct prediction sets for each selected individual, at the level of $\alpha|\text{BH}(\mathbf{q})|/m$, thereby providing FCR guarantee. To mitigate power loss from this adjustment process, they employ the method from Bao et al. [3] to select an initial subset $\mathcal{S}_0 \subset \mathcal{U}$ which reduces the number of units and allows for a larger adjusted level of $\alpha|\text{BH}|/|\mathcal{S}_0|$ to construct prediction sets. To maintain their theoretical guarantee, they require the initial selection to be joint-exchangeable, which connects to our setting.

The FCR guarantee of InfoSCOP can directly imply FDR control on a data-dependent selection set by ensuring that the "informative prediction set" is informative with respect to the null hypothesis being tested. In this way, InfoSCOP implements an FDR control procedure after selection by applying the BH procedure to selective conformal p-values, which closely aligns with the core approach of our work.

For strongly stable selections, our method can be simplified and degenerate into a form similar to InfoSCOP. But the assumption in InfoSCOP is not satisfied by the quantile selection rule based solely on test data. Thus, their theoretical results are not applicable in such cases, while our framework bridges this theoretical gap.

And our approach covers a wider range of selection rules. For instance, when dealing with weakly stable rules, we employ conditional calibration on adaptive $p$-values to ensure rigorous FDR guarantees. The table below compares the performance of our approach with InfoSCOP under mean selection rule. The InfoSCOP shows reasonable empirical performance, which is similar to ours. Therefore, it is possible that InfoSCOP may still work under mean selection, making it an interesting topic for theoretical investigation, which remains unexplored in InfoSCOP. In contrast, we provide FDR control guarantee under a variety of selection scenarios.

Table 4: Comparisons of empirical FDR (%) and Power (%) with target FDR level $\alpha = 10\%$ by 500 repetitions.

|  | CASE A | | CASE B | |
| --- | --- | --- | --- | --- |
|  | FDR | POWER | FDR | POWER |
| INFOSCOP | 9.85 | 94.0 | 9.80 | 78.4 |
| SCPV | 9.86 | 93.4 | 9.80 | 78.1 |

In conclusion, our approach and InfoSCOP are designed for different goals, resulting in different analytical frameworks. Ours is specifically designed to address the multiple testing problem across various selection rules. From the perspective of conditional calibration, our method is unified, where the BH procedure for strongly stable selection can be seen as a special case. As a comparison, InfoSCOP is an excellent work for selecting an informative set with FCR control, but it is not primarily designed for multiple testing after data-dependent selection. Their FDR guarantee is an extension of FCR control, which limits their method's applicability to different selection rules.

Table 5: Comparisons of empirical FDR (%) and Power (%) under different scenarios and thresholds with target FDR $\alpha = 10\%$ and noise strength $\sigma = 0.5$. The sample sizes of the labeled set and the test set are fixed as $n = m = 1200$.

| | CONSTANT | | EXCH | | TEST | |
|---|---|---|---|---|---|---|
| | FDR | POWER | FDR | POWER | FDR | POWER |
| **RANDOM FOREST** | | | | | | |
| SCPV | 9.81 | 95.27 | 9.78 | 95.23 | 9.80 | 95.23 |
| OMT | 19.07 | 98.87 | 19.07 | 98.87 | 19.07 | 98.87 |
| AMT(BH) | 6.14 | 92.41 | 6.11 | 92.35 | 6.14 | 92.36 |
| AMT(BY) | 0.73 | 79.20 | 0.72 | 79.15 | 0.71 | 79.13 |
| **SVM** | | | | | | |
| SCPV | 9.82 | 85.84 | 9.83 | 85.88 | 9.85 | 85.89 |
| OMT | 15.05 | 95.21 | 15.05 | 96.21 | 15.04 | 96.18 |
| AMT(BH) | 7.79 | 79.66 | 7.81 | 79.69 | 7.80 | 79.65 |
| AMT(BY) | 1.28 | 38.82 | 1.29 | 38.90 | 1.28 | 38.87 |
| **NEURALNET** | | | | | | |
| SCPV | 9.73 | 88.62 | 9.72 | 88.14 | 9.74 | 88.35 |
| OMT | 7.89 | 64.44 | 7.90 | 64.39 | 7.90 | 64.45 |
| AMT(BH) | 6.15 | 19.44 | 6.19 | 19.46 | 6.20 | 19.49 |
| AMT(BY) | 0.04 | 0.14 | 0.05 | 0.17 | 0.04 | 0.14 |

# E  Additional empirical results

## E.1  The effect of the learning models

In Table 5, we present the results of FDR and power under three machine learning methods. The data is generated based on the settings specified in case A. The neural network (Neuralnet) with a single hidden layer and 5 hidden neurons is implemented by using the R package `neuralnet`. And the linear output units are used. Here, we fix the noise strength $\sigma = 0.5$. It can be seen that our method controls FDR at the expected level and it also provides satisfactory testing power. In contrast, the FDR values obtained from AMT tend to be overly conservative, leading to a notable deflation of its power. In the first two settings, the OMT methods are inadequate in effectively controlling FDR. While they can successfully control FDR in the last setting, they often suffer from a loss of statistical power.

## E.2  A real data application with clustering

Diabetes is a chronic disease that affects a large and growing number of people worldwide [65]. As such, identifying potential diabetes patients using risk factors and machine learning tools is an attractive approach for early intervention and preventive measures. To this end, we applied our method to the Diabetes Health Indicators Dataset [38] provided by the Behavioral Risk Factor Surveillance System (BRFSS) in the United States. Through this analysis, we are able to effectively identify high-risk individuals while also providing uncertainty quantification measures.

In the dataset, the response variable is denoted as $Y_j$ which takes the value of 0 or 1, indicating whether the $j$-th person suffers from diabetes. The dataset also includes patient-related information consisting of 21 features, such as BMI (Body Mass Index), cholesterol level, and other health risk indicators. These covariates provide additional information about each individual that can be used to analyze and predict the likelihood of diabetes. Our goal is identifying those diabetes patients with controlled FDR $\alpha = 20\%$, i.e.

$$H_{0,j} : Y_j = 0 \quad \text{v.s.} \quad H_{1,j} : Y_j = 1.$$

The data is processed as follows: a total of $n = 2{,}000$ labeled data points and $m = 2{,}000$ test data points are randomly sampled from the dataset. The prediction model $\hat{\mu}$ is constructed by random forest using another $2{,}000$ i.i.d. training data. Based on prior knowledge, we understand that individuals with obesity are at a higher risk of suffering from type II diabetes [54]. Therefore, our focus is directed towards making inferences specifically on individuals with a high BMI. Denote the selected subset $\hat{\mathcal{S}}_u = \{j \in \mathcal{U} : T_j > \tau\}$. Several selection rules are considered. **Constant**: $T_i$ is the

BMI of $i$-th individual. $\tau = 30$. **Exch**: $T_i$ is the BMI of $i$-th individual and the $\tau$ is the $70\%$-quantile of $\{T_j : j \in \mathcal{C} \cup \mathcal{U}\}$. **Quan**: $T_i$ is the BMI of $i$-th individual. $\tau$ is the $70\%$-quantile of $\{T_j : j \in \mathcal{U}\}$.

Table 6 depicts the results of our proposed SCPV and other compared benchmarks using the thresholds mentioned above. Our method and adjusted methods successfully achieve valid FDR control using all of these selection rules. However, adjusted methods select fewer individuals, leading to powerless results. Meanwhile, the OMT fails to control FDR for most settings.

Table 6: Comparisons of empirical FDR (%) and Power (%) with target FDR level $\alpha = 20\%$ by 500 repetitions.

| | CONSTANT | | EXCH | | QUAN | |
|---|---|---|---|---|---|---|
| | FDR | POWER | FDR | POWER | FDR | POWER |
| SCPV | 19.99 | 63.99 | 20.01 | 72.43 | 20.03 | 72.26 |
| OMT | 23.88 | 82.99 | 23.14 | 87.55 | 23.18 | 87.44 |
| AMT(BH) | 13.21 | 23.52 | 8.58 | 12.54 | 8.66 | 12.93 |
| AMT(BY) | 0.03 | 0.05 | 0.02 | 0.05 | 0.02 | 0.04 |

Besides, the dataset is potentially composed of different groups, and it is important to identify individuals while controlling the FDR for each group. This allows us to make more accurate assessments and informed decisions specific to each group. To address this, we employ a clustering algorithm, specifically K-means, to divide the dataset (which uses both labeled and test data but lacks response information) into two distinct groups. Our primary objective is to draw inferences within each individual group. The clustering process results in two groups that exhibit significant disparities in terms of the "MentalHealth" covariate. Consequently, we refer to the group with a lower "MentalHealth" index as Group A, while the other group is denoted as Group B.

Regarding the results for the two clustered groups in Table 7, we observe that our method exhibits stringent FDR control. The OMT lacks power for Group A and yields an inflated FDR level for Group B. As for these adjusted multiple testing methods, they deliver more conservative rejection results. To conclude, our method is powerful to provide subgroup FDR control for adaptively chosen groups.

Table 7: Comparisons of empirical FDR (%) and Power (%) with target FDR level $\alpha = 20\%$ by 500 repetitions.

| | GROUP A | | GROUP B | |
|---|---|---|---|---|
| | FDR | POWER | FDR | POWER |
| SCPV | 16.35 | 12.75 | 19.79 | 77.05 |
| OMT | 4.58 | 1.43 | 24.04 | 95.60 |
| AMT(BH) | 1.84 | 0.38 | 11.54 | 23.96 |
| AMT(BY) | 0.00 | 0.00 | 0.17 | 0.16 |

### E.3 A real data application with deep learning method

Breast cancer is the most common form of cancer in women, with infiltrating ductal carcinoma (IDC) being the most common form of breast cancer. Accurately identifying and classifying subtypes of breast cancer is an important clinical task, and utilizing deep learning methods for identification can effectively save time and reduce errors. Our dataset consists of complete whole slide images of breast cancer (BCa) specimens scanned at 40 times magnification. Our method can effectively identify individuals who may be at risk of breast cancer, while also measuring the uncertainty of the deep learning model.

In this dataset, the label is denoted as Y, with Y taking values of 0 and 1, representing whether the j-th image is a slice from a breast cancer specimen. Our goal is identifying those breast cancer patients

with controlled FDR $\alpha = 10\%$, i.e.

$$H_{0,j} : Y_j = 0 \quad \text{v.s.} \quad H_{1,j} : Y_j = 1.$$

The data is processed as follows: a total of $n = 800$ labeled data points and $m = 800$ test data points are randomly sampled from the dataset. The prediction model $\hat{\mu}$ is constructed by a convolutional Neural Network with 10 layers using another 2000 i.i.d. training data. This network consists of a total of 10 layers, including 4 convolutional layers and 3 max pooling layers.

$T_i$ represents a score used to assess the risk of breast cancer, such as the probability of developing breast cancer predicted by a model. Our goal is to identify individuals in the high-risk group who are more likely to develop the disease. Denote the selected subset $\hat{\mathcal{S}}_u = \{j \in \mathcal{U} : T_j > \tau\}$. Several selection rules are considered. **Constant**: $T_i$ represents the predicted probability of the $i$-th individual obtained using model $\hat{\mu}$. $\tau = 0.2$. **Exch**: $T_i$ represents the predicted probability of the $i$-th individual obtained using model $\hat{\mu}$ and the $\tau$ is the 30%-quantile of $\{T_j : j \in \mathcal{C} \cup \mathcal{U}\}$. **Quan**: $T_i$ represents the predicted probability of the $i$-th individual obtained using model $\hat{\mu}$. $\tau$ is the 30%-quantile of $\{T_j : j \in \mathcal{U}\}$.

Table 8 displays the outcomes obtained by our proposed SCPV method and other benchmark approaches when employing the specified thresholds. Our method and adjusted methods successfully achieve valid FDR control under these criteria. However, adjusted methods select fewer individuals, resulting in powerless results. Meanwhile, the OMT proves inadequate in controlling FDR for most settings.

Table 8: Comparisons of empirical FDR (%) and Power (%) with target FDR level $\alpha = 10\%$ by 100 repetitions.

|  | **CONSTANT** | | **EXCH** | | **QUAN** | |
|---|---|---|---|---|---|---|
|  | FDR | POWER | FDR | POWER | FDR | POWER |
| SCPV | 9.88 | 72.9 | 9.71 | 74.0 | 9.75 | 74.1 |
| OMT | 14.1 | 80.3 | 16.9 | 88.3 | 17.0 | 88.7 |
| AMT(BH) | 9.24 | 71.4 | 9.24 | 73.3 | 9.24 | 73.6 |
| AMT(BY) | 0.479 | 9.43 | 0.479 | 9.71 | 0.479 | 9.80 |

# F    Technical proofs

## F.1    Revisit of notations

The index set of the labeled set and test set are $\mathcal{C}$ and $\mathcal{U}$. For any subset $\mathcal{S} \subseteq \mathcal{C} \cup \mathcal{U}$, we use $\mathcal{D}_{\mathcal{S}}$ to denote the data $\{i \in \mathcal{S} : (X_i, Y_i)\}$. The selected test set and calibration set are $\hat{\mathcal{S}}_u$ and $\hat{\mathcal{S}}_c$. The null labeled set and test set are denoted as $\mathcal{C}_0 = \{i \in \mathcal{C} : Y_i \notin \mathcal{A}\}$ and $\mathcal{U}_0 = \{j \in \mathcal{U} : Y_j \notin \mathcal{A}\}$. Equivalently, $\{Y_i\}$

## F.2    Auxiliary Lemmas

We introduce some auxiliary lemmas. The first one is the quantile inflation lemmas which is common in conformal inference literature [46, 3] and we omit its proof.

**Lemma F.1.** *Let* $\mathbf{x}_{(\lceil nt \rceil)}$ *is the* $\lceil nt \rceil$*-smallest value in* $\{\mathbf{x}_i \in \mathbb{R} : i \in [n]\}$*. Then for any* $t \in (0,1)$*, it holds that*

$$\frac{1}{n} \sum_{i=1}^{n} \mathbb{1}(\mathbf{x}_i \leq \mathbf{x}_{(\lceil nt \rceil)}) \leq t.$$

*If all values in* $\{\mathbf{x}_i : i \in [n]\}$ *are distinct, it also holds that*

$$\frac{1}{n} \sum_{i=1}^{n} \mathbb{1}(\mathbf{x}_i \leq \mathbf{x}_{(\lceil nt \rceil)}) \geq t - \frac{1}{n},$$

The next lemma is the key of our theoretical results, which characterizes the properties of the constructed p-values. The proof is deferred to Appendix F.4.

**Lemma F.2.** *If the data are i.i.d. and the selection rule $\mathbf{S}_{\mathcal{D}_c,\mathcal{D}_u}$ is strongly stable, we have that*

*(a) $\mathcal{R}^{(j\to 0)}$ defined in (9) is measurable with respect to $\Phi_j$ defined in 7.*

*(b) For any $j \in \mathcal{U}$, it holds that for any random variable $t \in \mathbb{R}$ that is measurable with respect to the unordered set $\Phi_j$, we have*

$$\mathbb{P}\left(p_j \leq t \mid j \in \hat{\mathcal{S}}_u, Y_j \notin \mathcal{A}, \Phi_j\right) \leq t.$$

### F.3 Proof of Theorem 3.2

*Proof.* We begin by establishing the validity of the selective conformal p-values. Consider the quantity

$$\Phi_j = (\mathcal{D}^*_{\mathcal{C}\cup\{j\}}, \mathcal{D}_{\mathcal{U}\setminus\{j\}}), \tag{7}$$

which consists of two components: $\mathcal{D}_{\mathcal{U}\setminus\{j\}}$, the test data with the $j$-th sample excluded, and $\mathcal{D}^*_{\mathcal{C}\cup\{j\}} := [Z_i; i \in \mathcal{C} \cup \{j\}]$, whose elements are taking values on $\{Z_i\}_{i\in\mathcal{C}} \cup \{Z_j\}$ but without their indexes. According to Lemma F.2(b), we have

$$\mathbb{P}\left(p_j \leq t \mid j \in \hat{\mathcal{S}}_u, Y_j \notin \mathcal{A}\right) = \mathbb{E}\left[\mathbb{E}\left[\mathbb{1}\{p_j \leq t\} \mid j \in \hat{\mathcal{S}}_u, Y_j \notin \mathcal{A}, \Phi_j\right]\right] \leq t$$

Next, we proceed to verify the control of FDR. This involves examining the previously defined defined p-values

$$p_j = \frac{1 + \sum_{i\in\hat{\mathcal{S}}_c\cap\mathcal{C}_0} \mathbb{1}\{V_i < V_j\}}{1 + |\hat{\mathcal{S}}_c \cap \mathcal{C}_0|}, \quad \text{for } j \in \hat{\mathcal{S}}_u.$$

For any $j \in \hat{\mathcal{S}}_u$, define a set of slightly modified p-values

$$p_\ell^{(j)} = \frac{\sum_{i\in\hat{\mathcal{S}}_c\cap\mathcal{C}_0} \mathbb{1}\{V_i < V_\ell\} + \mathbb{1}\{V_j < V_\ell\}}{1 + |\hat{\mathcal{S}}_c \cap \mathcal{C}_0|}, \quad \forall \ell \neq j, \quad \ell \in \hat{\mathcal{S}}_u. \tag{8}$$

These p-values are only used in our analysis. Also define $\mathcal{R}\left(\{a_j : j \in \hat{\mathcal{S}}_u\}\right) \subseteq \hat{\mathcal{S}}_u$ as the rejection (indices) set obtained by the BH procedure, from p-values taking on the values in $\{a_j : j \in \hat{\mathcal{S}}_u\}$.

In the sequel, we will compare $\mathcal{R}$ to

$$\mathcal{R}\left(\{p_l^{(j)} : \ell \neq j, \ell \in \hat{\mathcal{S}}_u\} \cup \{p_j\}\right)$$

on the event $\{Y_j \notin \mathcal{A}, j \in \mathcal{R}\}$. For the remaining p-values, since the scores have no ties, we consider two cases:

(i) If $V_j \leq V_\ell$, then

$$p_\ell^{(j)} = \frac{1 + \sum_{i\in\hat{\mathcal{S}}_c\cap\mathcal{C}_0} \mathbb{1}\{V_i < V_\ell\}}{|\hat{\mathcal{S}}_c \cap \mathcal{C}_0| + 1} = p_\ell.$$

(ii) If $V_j > V_\ell$, then $p_\ell \leq p_j$. Since $j \in \mathcal{R}$, the BH procedure implies $\ell \in \mathcal{R}$. By definition, we have

$$p_\ell^{(j)} \leq \frac{1 + \sum_{i\in\hat{\mathcal{S}}_c\cap\mathcal{C}_0} \mathbb{1}\{V_i < V_\ell\}}{1 + |\hat{\mathcal{S}}_c \cap \mathcal{C}_0|} \leq \frac{1 + \sum_{i\in\hat{\mathcal{S}}_c\cap\mathcal{C}_0} \mathbb{1}\{V_i < V_\ell\}}{1 + |\hat{\mathcal{S}}_c \cap \mathcal{C}_0|} = p_j.$$

To summarize, suppose we are to replace $p_\ell$ by $p_\ell^{(j)}$ for all $\ell \neq j, \ell \in \hat{\mathcal{S}}_u$. Then on the event $\{Y_j \notin \mathcal{A}, j \in \mathcal{R}\}$, such a replacement does not change any of those $p_\ell \geq p_j$; also, all those $p_\ell \leq p_j$ including $p_j$ itself (they are rejected in $\mathcal{R}$) are still no greater than $p_j$ after the replacement. Thus, by the step-up nature of the BH procedure, such a replacement does not change the rejection set, meaning that

$$\mathcal{R} = \mathcal{R}\left(\{p_j : j \in \hat{\mathcal{S}}_u\}\right)$$

$$= \mathcal{R}\left(\{p_\ell^{(j)} : \ell \neq j, \ell \in \hat{\mathcal{S}}_u\} \cup \{p_j\}\right) =: \mathcal{R}^{(j)}$$

on the event $\{Y_j \notin \mathcal{A}, j \in \mathcal{R}\}$. Let $\mathbb{R}_j = \mathbb{1}\{j \in \mathbb{R}\}$, then a leave-one-out analysis of the FDR implies

$$\text{FDR} = \mathbb{E}\left[\frac{\sum_{j \in \hat{\mathcal{S}}_u} \mathbb{1}\{Y_j \notin \mathcal{A}\} R_j}{1 \vee \sum_{j \in \hat{\mathcal{S}}_u} R_j}\right]$$

$$\stackrel{(i)}{=} \sum_{j \in \mathcal{U}} \mathbb{E}\left[\sum_{k=1}^{|\hat{\mathcal{S}}_u|} \frac{1}{k} \mathbb{1}\{|\mathcal{R}| = k\} \mathbb{1}\{Y_j \notin \mathcal{A}\} \mathbb{1}\left\{p_j \leq \alpha k/|\hat{\mathcal{S}}_u|\right\} \mathbb{1}\{j \in \hat{\mathcal{S}}_u\}\right]$$

$$\stackrel{(ii)}{=} \sum_{j \in \mathcal{U}} \mathbb{E}\left[\sum_{k=1}^{|\hat{\mathcal{S}}_u|} \frac{1}{k} \mathbb{1}\left\{\left|\mathcal{R}^{(j)}\right| = k\right\} \mathbb{1}\{Y_j \notin \mathcal{A}\} \mathbb{1}\left\{j \in \mathcal{R}^{(j)}\right\} \mathbb{1}\{j \in \hat{\mathcal{S}}_u\}\right].$$

The $(i)$ use the property of the BH procedure, and $(ii)$ comes from the facts stated just above. By the step-up nature of the BH procedure, we know that on the event $\{j \in \mathcal{R}^{(j)}\}$, sending $p_j$ to zero does not change the rejection set, i.e., we have

$$\mathcal{R}^{(j)} = \mathcal{R}\left(\{p_\ell^{(j)} : \ell \neq j, \ell \in \hat{\mathcal{S}}_u\} \cup \{0\}\right) =: \mathcal{R}^{(j \to 0)}. \tag{9}$$

Thus

$$\text{FDR} = \sum_{j \in \mathcal{U}} \mathbb{E}\left[\sum_{k=1}^{|\hat{\mathcal{S}}_u|} \frac{1}{k} \mathbb{1}\left\{\left|\mathcal{R}^{(j \to 0)}\right| = k\right\} \mathbb{1}\left\{p_j \leq \alpha |\mathcal{R}^{(j \to 0)}|/|\hat{\mathcal{S}}_u|\right\} \mathbb{1}\{Y_j \notin \mathcal{A}\} \mathbb{1}\{j \in \hat{\mathcal{S}}_u\}\right] \tag{10}$$

$$= \sum_{j \in \mathcal{U}} \mathbb{E}\left[\frac{\mathbb{1}\left\{p_j \leq \alpha \left|\mathcal{R}^{(j \to 0)}\right|/|\hat{\mathcal{S}}_u|\right\} \mathbb{1}\{Y_j \notin \mathcal{A}\} \mathbb{1}\{j \in \hat{\mathcal{S}}_u\}}{1 \vee \left|\mathcal{R}^{(j \to 0)}\right|}\right] \tag{11}$$

By Lemma F.2(a), it holds that

$$\mathbb{E}\left[\frac{\mathbb{1}\left\{p_j \leq \alpha \left|\mathcal{R}^{(j \to 0)}\right|/|\hat{\mathcal{S}}_u|\right\} \mathbb{1}\{Y_j \notin \mathcal{A}\} \mathbb{1}\{j \in \hat{\mathcal{S}}_u\}}{1 \vee \left|\mathcal{R}^{(j \to 0)}\right|} \mid \Phi_j\right]$$

$$= \frac{1}{1 \vee \left|\mathcal{R}^{(j \to 0)}\right|} \mathbb{E}\left[\mathbb{1}\left\{p_j \leq \alpha \left|\mathcal{R}^{(j \to 0)}\right|/|\hat{\mathcal{S}}_u|\right\} \mathbb{1}\{Y_j \notin \mathcal{A}\} \mathbb{1}\{j \in \hat{\mathcal{S}}_u\} \mid \Phi_j\right]$$

$$= \frac{1}{1 \vee \left|\mathcal{R}^{(j \to 0)}\right|} \mathbb{P}\left(p_j \leq \alpha \left|\mathcal{R}^{(j \to 0)}\right|/|\hat{\mathcal{S}}_u| \mid j \in \hat{\mathcal{S}}_u, Y_j \notin \mathcal{A}, [\{V_i\}_{i \in \mathcal{C} \cup \{j\}}]\right) \mathbb{P}\left(j \in \hat{\mathcal{S}}_u, Y_j \notin \mathcal{A} \mid \Phi_j\right)$$

and $\left|\mathcal{R}^{(j \to 0)}\right|/|\hat{\mathcal{S}}_u|$ is measurable with respect to the unordered set $\Phi_j$. Then use Lemma F.2(b), we have

$$\mathbb{P}\left(p_j \leq \alpha \left|\mathcal{R}^{(j \to 0)}\right|/|\hat{\mathcal{S}}_u| \mid j \in \hat{\mathcal{S}}_u, Y_j \notin \mathcal{A}, \Phi_j\right) \leq \alpha \left|\mathcal{R}^{(j \to 0)}\right|/|\hat{\mathcal{S}}_u|.$$

Through summing over $j \in \mathcal{U}$, together with tower's rule, this gives

$$\text{FDR} \leq \sum_{j \in \mathcal{U}} \alpha \mathbb{E}\left[\frac{\mathbb{1}\{j \in \hat{\mathcal{S}}_u, Y_j \notin \mathcal{A}\}}{|\hat{\mathcal{S}}_u|}\right] = \alpha \mathbb{E}\left[\frac{|\hat{\mathcal{S}}_u \cap \mathcal{U}_0|}{|\hat{\mathcal{S}}_u|}\right] \leq \alpha$$

which concludes the proof. □

### F.4 Proof of Lemma F.2

*Proof.* We note that the following proof is conditioned on the event $\{Y_j \notin \mathcal{A}\}$. Define

$$\hat{\mathcal{S}}_{c_0, +j} = \{i \in \mathcal{C} \cup \{j\} : \mathbf{S}_{\mathcal{D}_c, \mathcal{D}_u}(X_j) = 1, y_i \notin \mathcal{A}\}.$$

Then $\hat{\mathcal{S}}_c \cap \mathcal{C}_0 \cup \{j\} = \hat{\mathcal{S}}_{c_0,+j}$ and $|\hat{\mathcal{S}}_c \cap \mathcal{C}_0| + 1 = |\hat{\mathcal{S}}_{c_0,+j}|$ hold under the event $j \in \hat{\mathcal{S}}_u$ and $Y_j \notin \mathcal{A}$. For any $j$ satisfying $Y_j \notin \mathcal{A}$, define the event

$$\mathcal{A}_{\mathcal{C}\cup\{j\}}(z) = \{[Z_{i\in\mathcal{C}\cup\{j\}}] = [z_1, \cdots, z_n, z_{n+1}]\}. \tag{12}$$

Denote the corresponding unordered conformal scores by $[v_1, \cdots, v_{n+1}]$ under $\mathcal{A}_{\mathcal{C}\cup\{j\}}(z)$. Since $\mathbf{S}_{\mathcal{D}_c, \mathcal{D}_u}$ is strongly stable, then given $D_{\mathcal{U}\setminus j}$ and under $\mathcal{A}_{\mathcal{C}\cup\{j\}}(z)$ we have

$$\mathbf{S}_{\mathcal{D}_c, \mathcal{D}_u}(x_i) = \mathbf{S}_{\mathcal{D}_c\cup\{Z_j\}, \mathcal{D}_u\setminus\{Z_j\}}(x_i) = \mathbf{S}_{[z_1, \cdots, z_n, z_{n+1}], \mathcal{D}_u\setminus\{Z_j\}}(x_i)$$

for $i = 1, \cdots, n+1$. It means that the following unordered set

$$\left[\{V_i\}_{i\in\hat{\mathcal{S}}_{c_0,+j}}\right] \mid \mathcal{A}_{\mathcal{C}\cup\{j\}}(z) = \left[\{v_i : \mathbf{S}_{z, \mathcal{D}_u\setminus\{Z_j\}}(x_i) = 1, y_i \notin \mathcal{A}\}_{i=1,\cdots,n+1}\right]$$

is known and only depend on $z$ and the data $D_{\mathcal{U}\setminus j}$. Besides,

$$|\hat{\mathcal{S}}_{c_0,+j}| \mid \mathcal{A}_{\mathcal{C}\cup\{j\}}(z) = |\{i \in \{1, \cdots, n+1\} : \mathbf{S}_{z, \mathcal{D}_u\setminus\{Z_j\}}(x_i) = 1, y_i \notin \mathcal{A}\}|$$

is also known. Note that by definition (8), $\left\{p_\ell^{(j)}\right\}_{\ell\neq j}$ is invariant after permuting $\{V_i\}_{i\in\hat{\mathcal{S}}_c\cap\mathcal{C}_0} \cup \{V_j\}$. We know that the modified $p$-value

$$p_\ell^{(j)} \mid \mathcal{A}_{\mathcal{C}\cup\{j\}}(z) = \frac{\sum_{i\in\hat{\mathcal{S}}_{c_0,+j}} \mathbb{1}\{v_i < V_\ell\}}{|\hat{\mathcal{S}}_{c_0,+j}|}, \quad \forall \ell \neq j, \quad \ell \in \hat{\mathcal{S}}_u.$$

is fixed condition on $\mathcal{A}_{\mathcal{C}\cup\{j\}}(z)$. Also note that $\mathcal{R}^{(j\to 0)}$ only depends on $\left\{p_j^{(\ell)}\right\}_{\ell\neq j}$, and this implies that $\mathcal{R}^{(j\to 0)}$ is known under $\mathcal{A}_{\mathcal{C}\cup\{j\}}(z)$. Through marginalizing over $\mathcal{A}_{\mathcal{C}\cup\{j\}}(z)$, we obtain that $p_\ell^{(j)}$ is measurable with respect to $\Phi_j$. Since $\mathcal{R}^{(j\to 0)}$ is only depend on $p_\ell^{(j)}$ for $\ell \neq j$, $\ell \in \hat{\mathcal{S}}_u$, the first part of Lemma F.2 can be readily demonstrated.

For the second part, we know it holds that

$$\{p_j \leq t\} = \{V_j \leq \mathbf{V}_{(\lceil t(|\hat{\mathcal{S}}_c\cap\mathcal{C}_0|+1)\rceil)}^{\hat{\mathcal{S}}_c\cap\mathcal{C}_0}\} = \{V_j \leq \mathbf{V}_{(\lceil t(|\hat{\mathcal{S}}_c\cap\mathcal{C}_0|+1)\rceil)}^{\hat{\mathcal{S}}_c\cap\mathcal{C}_0\cup\{j\}}\}$$
$$= \{V_j \leq \mathbf{V}_{(\lceil t|\hat{\mathcal{S}}_{c_0,+j}(\hat{\tau})|\rceil)}^{|\hat{\mathcal{S}}_{c_0,+j}(\hat{\tau})|}\}$$

by the construction of $p_j$ and Lemma F.1. And then we have

$$\mathbb{P}\left(p_j \leq t, Y_j \notin \mathcal{A} \mid j \in \hat{\mathcal{S}}_u, \Phi_j\right)$$
$$=\mathbb{P}\left(V_j \leq \mathbf{V}_{(\lceil t(|\hat{\mathcal{S}}_c\cap\mathcal{C}_0|+1)\rceil)}^{\hat{\mathcal{S}}_c\cap\mathcal{C}_0\cup\{j\}}, Y_j \notin \mathcal{A} \mid \mathbf{S}_{\mathcal{D}_c,\mathcal{D}_u}(X_j) = 1, \Phi_j\right)$$
$$\leq t + \mathbb{E}\left[\frac{1}{|\hat{\mathcal{S}}_c\cap\mathcal{C}_0|+1} \sum_{k\in\hat{\mathcal{S}}_c\cap\mathcal{C}_0} \mathbb{1}\left\{V_j \leq \mathbf{V}_{(\lceil t(|\hat{\mathcal{S}}_c\cap\mathcal{C}_0|+1)\rceil)}^{\hat{\mathcal{S}}_c\cap\mathcal{C}_0\cup\{j\}}, Y_j \notin \mathcal{A}\right\}\right.$$
$$\left. -\mathbb{1}\left\{V_k \leq \mathbf{V}_{(\lceil t(|\hat{\mathcal{S}}_c\cap\mathcal{C}_0|+1)\rceil)}^{\hat{\mathcal{S}}_c\cap\mathcal{C}_0\cup\{j\}}, Y_j \notin \mathcal{A}\right\} \mid \mathbf{S}_{\mathcal{D}_c,\mathcal{D}_u}(X_j) = 1, \Phi_j\right]$$
$$=t + \sum_{k\in\mathcal{C}} \mathbb{E}\left[\frac{1}{|\hat{\mathcal{S}}_c\cap\mathcal{C}_0|+1} \mathbb{1}\left\{V_j \leq \mathbf{V}_{(\lceil t(|\hat{\mathcal{S}}_c\cap\mathcal{C}_0|+1)\rceil)}^{\hat{\mathcal{S}}_c\cap\mathcal{C}_0\cup\{j\}}, Y_j \notin \mathcal{A}, \mathbf{S}_{\mathcal{D}_c,\mathcal{D}_u}(X_k) = 1, Y_k \notin \mathcal{A}\right\}\right.$$
$$\left. -\frac{1}{|\hat{\mathcal{S}}_c\cap\mathcal{C}_0|+1} \mathbb{1}\left\{V_k \leq \mathbf{V}_{(\lceil t(|\hat{\mathcal{S}}_c\cap\mathcal{C}_0|+1)\rceil)}^{\hat{\mathcal{S}}_c\cap\mathcal{C}_0\cup\{j\}}, Y_j \notin \mathcal{A}, \mathbf{S}_{\mathcal{D}_c,\mathcal{D}_u}(X_k) = 1, Y_k \notin \mathcal{A}\right\} \mid \mathbf{S}_{\mathcal{D}_c,\mathcal{D}_u}(X_j) = 1, \Phi_j\right]$$
$$=t + \sum_{k\in\mathcal{C}} \frac{1}{\mathbb{P}(\mathbf{S}_{\mathcal{D}_c,\mathcal{D}_u}(X_j) = 1)} \mathbb{E}\left[\frac{1}{|\hat{\mathcal{S}}_{c_0,+j}|} \mathbb{1}\left\{V_j \leq \mathbf{V}_{(\lceil t(|\hat{\mathcal{S}}_{c_0,+j}|)\rceil)}^{\hat{\mathcal{S}}_{c_0,+j}}, \mathbf{S}_{\mathcal{D}_c,\mathcal{D}_u}(X_j) = 1, Y_j \notin \mathcal{A}, \mathbf{S}_{\mathcal{D}_c,\mathcal{D}_u}(X_k) = 1, Y_k \notin \mathcal{A}\right\}\right.$$
$$\left. -\frac{1}{|\hat{\mathcal{S}}_{c_0,+j}|} \mathbb{1}\left\{V_k \leq \mathbf{V}_{(\lceil t(|\hat{\mathcal{S}}_{c_0,+j}|)\rceil)}^{\hat{\mathcal{S}}_{c_0,+j}}, \mathbf{S}_{\mathcal{D}_c,\mathcal{D}_u}(X_j) = 1, Y_j \notin \mathcal{A}, \mathbf{S}_{\mathcal{D}_c,\mathcal{D}_u}(X_k) = 1, Y_k \notin \mathcal{A}\right\} \mid \Phi_j\right]$$

$$\tag{13}$$

For ease of presentation, we write

$$|\hat{\mathcal{S}}_{c_0,+j}| \mid \mathcal{A}_{\mathcal{C}\cup\{j\}}(z) = \mathcal{S}_{c_0,+j}(z; D_{\mathcal{U}\backslash j}),$$

$$\mathbf{V}^{\hat{\mathcal{S}}_{c_0,+j}}_{(\lceil t(|\hat{\mathcal{S}}_{c_0,+j}|)\rceil)} \mid \mathcal{A}_{\mathcal{C}\cup\{j\}}(z) = \mathbf{V}(z; D_{\mathcal{U}\backslash j}).$$

For any unordered set $z$, we define the following unordered set

$$\Omega(z) = \big((i_1, i_2) \subseteq [n+1] : v_{i_1} \le \mathbf{V}(z; D_{\mathcal{U}\backslash j}), \mathbf{S}_{z,\mathcal{D}_u\backslash\{Z_j\}}(x_{i_1}) = 1, y_{i_1} \notin \mathcal{A}, \mathbf{S}_{z,\mathcal{D}_u\backslash\{Z_j\}}(x_{i_2}) = 1, y_{i_2} \notin \mathcal{A}\big)$$

which is $\sigma(\mathcal{D}_{\mathcal{U}\backslash j})$-measurable and independent of $\mathcal{A}_{\mathcal{C}\cup\{j\}}(z)$. Using the exchangeability of $(Z_i)_{i\in\mathcal{C}\cup\mathcal{U}}$, we can guarantee

$$\mathbb{E}\left[\frac{\mathcal{S}_{c_0,+j}(z; D_{\mathcal{U}\backslash j})}{|\hat{\mathcal{S}}_{c_0,+j}|}\mathbb{1}\left\{V_j \le \mathbf{V}^{\hat{\mathcal{S}}_{c_0,+j}}_{(\lceil t(|\hat{\mathcal{S}}_{c_0,+j}|)\rceil)}, \mathbf{S}_{\mathcal{D}_c,\mathcal{D}_u}(X_j) = 1, Y_j \notin \mathcal{A}, \mathbf{S}_{\mathcal{D}_c,\mathcal{D}_u}(X_k) = 1, Y_k \notin \mathcal{A}\right\} \mid \mathcal{A}_{\mathcal{C}\cup\{j\}}(z)\right]$$

$$=\mathbb{E}\left[\mathbb{1}\left\{V_j \le \mathbf{V}(z; D_{\mathcal{U}\backslash j}), \mathbf{S}_{z,\mathcal{D}_u\backslash\{Z_j\}}(X_j) = 1, Y_j \notin \mathcal{A}, \mathbf{S}_{z,\mathcal{D}_u\backslash\{Z_j\}}(X_k) = 1, Y_k \notin \mathcal{A}\right\} \mid \mathcal{A}_{\mathcal{C}\cup\{j\}}(z)\right]$$

$$= \sum_{(i_1,i_2)\subseteq\Omega(z)} \mathbb{P}\left\{Z_j = z_{i_1}, Z_k = z_{i_2} \mid \mathcal{A}_{\mathcal{C}\cup\{j\}}(z)\right\}$$

$$= \sum_{(i_1,i_2)\subseteq\Omega(z)} \mathbb{P}\left\{Z_j = z_{i_2}, Z_k = z_{i_1} \mid \mathcal{A}_{\mathcal{C}\cup\{j\}}(z)\right\}$$

$$=\mathbb{E}\left[\mathbb{1}\left\{V_k \le \mathbf{V}(z; D_{\mathcal{U}\backslash j}), \mathbf{S}_{z,\mathcal{D}_u\backslash\{Z_j\}}(X_k) = 1, Y_k \notin \mathcal{A}, \mathbf{S}_{z,\mathcal{D}_u\backslash\{Z_j\}}(X_j) = 1, Y_j \notin \mathcal{A}\right\} \mid \mathcal{A}_{\mathcal{C}\cup\{j\}}(z)\right]$$

Through marginalizing over $\mathcal{A}_{\mathcal{C}\cup\{j\}}(z)$, it follows that

$$\mathbb{E}\left[\frac{1}{|\hat{\mathcal{S}}_{c_0,+j}|}\mathbb{1}\left\{V_j \le \mathbf{V}^{\hat{\mathcal{S}}_{c_0,+j}}_{(\lceil t(|\hat{\mathcal{S}}_{c_0,+j}|)\rceil)}, \mathbf{S}_{z,\mathcal{D}_u\backslash\{Z_j\}}(X_j) = 1, Y_j \notin \mathcal{A}, \mathbf{S}_{z,\mathcal{D}_u\backslash\{Z_j\}}(X_k) = 1, Y_k \notin \mathcal{A}\right\} \mid \Phi_j\right]$$

$$=\mathbb{E}\left[\frac{1}{|\hat{\mathcal{S}}_{c_0,+j}|}\mathbb{1}\left\{V_k \le \mathbf{V}^{\hat{\mathcal{S}}_{c_0,+j}}_{(\lceil t(|\hat{\mathcal{S}}_{c_0,+j}|)\rceil)}, \mathbf{S}_{z,\mathcal{D}_u\backslash\{Z_j\}}(X_j) = 1, Y_j \notin \mathcal{A}, \mathbf{S}_{z,\mathcal{D}_u\backslash\{Z_j\}}(X_k) = 1, Y_k \notin \mathcal{A}\right\} \mid \Phi_j\right]$$

Plug into (13), we can verify the second part of Lemma F.2 immediately. $\qquad\square$

### F.5 Proof of Proposition 3.4

*Proof.* The results are direct if we let $\mathcal{D}_k = \mathcal{D}_c \cup \{Z_j\}$ and $\mathcal{D}_l = \mathcal{D}_u \backslash \{Z_j\}$ as a specific partition of $\mathcal{D}_c \cup \mathcal{D}_u$. By the definition of joint-exchangeable selection, we have

$$\mathbf{S}_{\mathcal{D}_c,\mathcal{D}_u}(X_i) = \mathbf{S}_{\mathcal{D}_k,\mathcal{D}_l}(X_i) = \mathbf{S}_{\mathcal{D}_c\cup\{Z_j\},\mathcal{D}_u\backslash\{Z_j\}}(X_i),$$

for any $i \in \mathcal{C} \cup \mathcal{U}$ and $j \in \mathcal{U}$. Thus the proof is completed. $\qquad\square$

### F.6 Proof of Proposition 3.5

*Proof.* Let $\{T_{(r)} : r \in [m]\}$ be order statistics of $\{T_{n+1}, \cdots, T_{j-1}, T_j, T_{j+1}, \cdots, T_{n+m}\}$ and $\{T^{j\to-\infty}_{(r)} : r \in [m]\}$ be order statistics of $\{T_{n+1}, \cdots, T_{j-1}, -\infty, T_{j+1}, \cdots, T_{n+m}\}$. By definition,

$$\tau_{\mathrm{topK}}(T_{n+1}, \cdots, T_{j-1}, T_j, T_{j+1}, \cdots, T_{n+m}) = T_{(K+1)}$$

and

$$\tau_{\mathrm{topK}}(T_{n+1}, \cdots, T_{j-1}, -\infty, T_{j+1}, \cdots, T_{n+m}) = T^{j\to-\infty}_{(K+1)}.$$

Note that for $j \in \hat{\mathcal{S}}_u$, we have $T_{(K+1)} > T_j$. Because $T_{(r)} = T^{j\to-\infty}_{(r)}$ for all order statistics with $T_{(r)} \ge T_j$, we have $T_{(K+1)} = T^{j\to-\infty}_{(K+1)}$ as well.

Therefore, we replace $Z_j = z$ for any $j \in \hat{\mathcal{S}}_u$, where $z$ is a fixed value such that the corresponding selection score $T_j$ (determined by the covariates $X_j$) is $-\infty$. Here we address that the top-K selection does not the specific scale of the selection scores. Hence the we can scale the selection scores into the range of $(0,1)$, and denote $z$ as the value such that $T_j = 0$.

As the threshold $\tau_{\mathrm{topK}}$ keeps unchanged after replacing $Z_j$ as $z$, we have

$$\mathbf{S}_{\mathcal{D}_c,\mathcal{D}_u}(X_i) = \mathbf{S}_{\mathcal{D}_c,\mathcal{D}_u\backslash\{Z_j\}\cup\{z\}}(X_i)$$

for any $j \in \hat{\mathcal{S}}_u$ and $i \in \mathcal{C} \cup \mathcal{U}$. $\qquad\square$

## F.7 Proof of Proposition 3.7

*Proof.* For simplicity, we suppose that the selection rule produces a selection threshold $\tau(\{T_k\}_{k\in\mathcal{U}})$ which is dependent on test data only. The selected test set is denoted as $\hat{\mathcal{S}}_u = \{j \in \mathcal{U} : T_j \leq \tau(\{T_k\}_{k\in\mathcal{U}})\}$. And the calibration set is picked up the by

$$\hat{\mathcal{S}}_c(j) = \{i \in \mathcal{C} : T_i \leq \tau(\{T_k\}_{k\in\mathcal{U}\setminus\{j\}})\} \text{ for } j \in \hat{\mathcal{S}}_u.$$

We note that the proof is similar for all type of weakly stable selection rule. Given $\mathcal{D}_u\setminus\{j\}$, we know that $\tau(\{T_k\}_{k\in\mathcal{U}\setminus\{j\}})$ is fixed. Therefore, for $j \in \hat{\mathcal{S}}_u$, i.e. $T_j \leq \tau(\{T_k\}_{k\in\mathcal{U}\setminus\{j\}})$ , it holds that $\{Z_i \in \mathcal{D}_c : T_i \leq \tau(\{T_k\}_{k\in\mathcal{U}\setminus\{j\}})\} = \{Z_i : i \in \hat{\mathcal{S}}_c(j)\}$ and $Z_j$ are exchangeable. Consequently, for $j \in \mathcal{U}_0$ and $j \in \hat{\mathcal{S}}_u$, it follows that $\{Z_i : i \in \hat{\mathcal{S}}_c(j)\cap\mathcal{C}_0\}$ and $Z_j$ are exchangeable, which implies $\{V_i : i \in \hat{\mathcal{S}}_c(j)\cap\mathcal{C}_0\}$ and $V_j$ are also exchangeable.

By the definition of $p_j^{adapt}$ and the similar procedure for proving Lemma F.2 in Appendix F.4, it is direct that $\Pr(p_j^{\text{adapt}} \leq t \mid j \in \hat{\mathcal{S}}_u, j \in \mathcal{U}_0) \leq t$. $\qquad\square$

## F.8 Proof of Theorem B.1

*Proof.* Our proof follows the same strategy as [22, 34]. Recall the final rejection set by conditional calibration can be formulated by

$$\mathcal{R} = \{j \in \mathcal{R}_+ : \epsilon_j \leq \frac{R}{\hat{R}_j}\},$$

where $R = |\mathcal{R}|$, $\hat{R}_j$ is the rejection number by BH procedure applied to $\{p_\ell^{\text{adapt},(j)} : \ell \neq j, \ell \in \hat{\mathcal{S}}_u\} \cup \{0\}$ and

$$\mathcal{R}_+ = \{j \in \hat{\mathcal{S}}_u : p_j^{\text{adapt}} \leq \frac{\alpha\hat{R}_j}{|\hat{\mathcal{S}}_u|}\}.$$

Define $\epsilon_{-j}$ as all $\epsilon_i$ variables for $i \in \mathcal{R}_+ \setminus \{j\}$ and $R^* = R(\epsilon_j \leftarrow 0)$ denote the hypothetical total number of rejections obtained by fixing $\epsilon_j = 0$ prior to applying conditional calibration procedure. Then, the FDR can be written as

$$
\begin{aligned}
\text{FDR} &= \sum_{j\in\mathcal{U}} \mathbb{E}\left[\frac{\mathbb{1}\{j \in \mathcal{R}_+\}\mathbb{1}\{\epsilon_j \leq \frac{R}{\hat{R}_j}\}\mathbb{1}\{Y_j \notin \mathcal{A}\}}{1 \vee R}\right] \\
&\stackrel{(i)}{=} \sum_{j\in\mathcal{U}} \mathbb{E}\left[\frac{\mathbb{1}\{j \in \mathcal{R}_+\}\mathbb{1}\{\epsilon_j \leq \frac{R^*}{\hat{R}_j}\}\mathbb{1}\{Y_j \notin \mathcal{A}\}}{1 \vee R^*}\right] \\
&= \sum_{j\in\mathcal{U}} \mathbb{E}\left[\mathbb{E}\left[\frac{\mathbb{1}\{j \in \mathcal{R}_+\}\mathbb{1}\{\epsilon_j \leq \frac{R^*}{\hat{R}_j}\}\mathbb{1}\{Y_j \notin \mathcal{A}\}}{1 \vee R^*}\right] \Big| \epsilon_{-j}, \mathcal{D}_c \cup \mathcal{D}_u\right] \\
&\stackrel{(ii)}{=} \sum_{j\in\mathcal{U}} \mathbb{E}\left[\frac{\mathbb{1}\{j \in \mathcal{R}_+\}\mathbb{1}\{Y_j \notin \mathcal{A}\}}{1 \vee \hat{R}_j}\right] \\
&= \sum_{j\in\mathcal{U}} \mathbb{E}\left[\frac{\mathbb{1}\{j \in \hat{\mathcal{S}}_u\}\mathbb{1}\{p_j^{\text{adapt}} \leq \frac{\alpha\hat{R}_j}{|\hat{\mathcal{S}}_u|}\}\mathbb{1}\{Y_j \notin \mathcal{A}\}}{1 \vee \hat{R}_j}\right] \\
&\stackrel{(iii)}{=} \sum_{j\in\mathcal{U}} \mathbb{E}\left[\frac{\mathbb{E}\left[\mathbb{1}\{j \in \hat{\mathcal{S}}_u\}\mathbb{1}\{p_j^{\text{adapt}} \leq \frac{\alpha\hat{R}_j}{|\hat{\mathcal{S}}_u|}\}\mathbb{1}\{Y_j \notin \mathcal{A}\} \mid \Phi_j\right]}{1 \vee \hat{R}_j}\right] \\
&\stackrel{(vi)}{\leq} \sum_{j\in\mathcal{U}} \mathbb{E}\left[\frac{\alpha\hat{R}_j}{|\hat{\mathcal{S}}_u|}\frac{\mathbb{1}\{j \in \hat{\mathcal{S}}_u\}\mathbb{1}\{Y_j \notin \mathcal{A}\}}{1 \vee \hat{R}_j}\right]
\end{aligned}
$$

$$= \alpha \sum_{j \in \mathcal{U}} \mathbb{E} \left[ \frac{\mathbb{1}\{j \in \hat{\mathcal{S}}_u\} \mathbb{1}\{Y_j \notin \mathcal{A}\}}{|\hat{\mathcal{S}}_u|} \right]$$

$$= \alpha \mathbb{E} \left[ \frac{|\hat{\mathcal{S}}_u \cap \mathcal{U}_0|}{|\hat{\mathcal{S}}_u|} \right].$$

Equality (i) holds since $\mathcal{R} = \mathcal{R}^*$ for $j \in \hat{\mathcal{S}}_u$, as the pruning procedure can be seen as a special BH procedure, which is not influence by replacing a rejected p-value with $0$. Equality (ii) is true because $\epsilon_j$ is independent of $\epsilon_{-j}$ given all the data $\mathcal{D}_c \cup \mathcal{D}_u$, and $\mathcal{R}^*$ is measurable to $\epsilon_{-j}, \mathcal{D}_c \cup \mathcal{D}_u$, where $\epsilon_j$ has no influence on $\mathcal{R}^*$ by the assignment ($\epsilon_j \leftarrow 0$). Equality (iii) holds as $\hat{\mathcal{R}}_j$ is measurable with respect to $\Phi_j$ by the design of $p_l^{\mathrm{adapt},(j)}$. And inequality (iv) comes from the Proposition 3.7, since the weak stability implies $|\hat{\mathcal{S}}_u| = |\{i \in \mathcal{U} : T_i < \tau(\{T_k\}_{k \in \mathcal{U} \setminus \{j\}})\}|$ is measurable with respect to $\Phi_j$ and $j \in \hat{\mathcal{S}}_u$. Thus the proof is completed. $\qquad \square$

