# OpenReview forum: "Conformalized Multiple Testing after Data-dependent Selection"
_NeurIPS.cc/2024/Conference — NeurIPS 2024 poster_

### Official Review · Reviewer_SifE · 2024-07-08

**Soundness:** 3
**Presentation:** 2
**Contribution:** 2
**Rating:** 6
**Confidence:** 3

**Summary:**

This paper addresses the problem of conformalized multiple testing following data-dependent selection procedures. To manage the distorted distribution resulting from the selection process, the authors propose adapting the calibration set according to the selection rule. Under the assumption that the selection procedure is stable or weakly stable, the authors prove that the guarantee of the Benjamini-Hochberg procedure is still maintained. Experiments on synthetic and real data have shown the effectiveness and efficiency of the method.

**Strengths:**

1. Multiple testing after data-dependent selection in the predictive setting is an important problem.
2. The definition of selection stability and weak stability is formalized and generalized to more selection conditions. The authors provide extensive theoretical proof for the theorem.
3. Adequate experiments demonstrate the effectiveness and efficiency of the proposed method.

**Weaknesses:**

1. The methodological novelty is limited: the idea of adapting the calibration set with the selection strategy to retain exchangeability is not very novel. Although dealing with conformal prediction, [1] also constructs the reference set based on the selection rule. The investigated selection methods are also similar. Overall, the main contribution of this method may be the rigorous proof (which, due to the length, I was unable to verify in every detail within the time constraints).

2. Minor errors:

    a. Line 28, $D_u = \{Z_i\}$ should be $D_u = \{X_i\}$.

    b. Line 153, 'label set' should be 'labeled set'.

[1] Ying Jin and Zhimei Ren. Confidence on the Focal: Conformal Prediction with Selection Conditional Coverage.

**Questions:**

May I ask if the idea of adapting the calibration set is similar to constructing the reference set in [1]?

**Limitations:**

The authors have adequately addressed the limitations of the i.i.d. setting and noted that the stability assumptions of the selection procedures are relatively limited.

---

> ### Author Rebuttal · Authors · 2024-08-06
>
> > Weakness 1: The methodological novelty is limited: the idea of adapting the calibration set with the selection strategy to retain exchangeability is not very novel. Although dealing with conformal prediction, [1] also constructs the reference set based on the selection rule. The investigated selection methods are also similar. Overall, the main contribution of this method may be the rigorous proof (which, due to the length, I was unable to verify in every detail within the time constraints).
>
> **To W1**: Thanks for your insightful comments on the methodological novelty of our approach. We agree that recent research has explored selective p-values from various perspectives. However, our work differs significantly from these studies, and we would like to provide some clarifications.
>
> - The references you mentioned have made valuable contributions by studying the properties of selective p-values, with a focus on individual p-values.  However, **our paper addresses multiple testing**, which requires studying the **interactions among all selective p-values rather than focusing on a single p-value**. This distinguishes our work from previous works in the literature. Specially, in our selective setting, the p-values are complicatedly correlated and the selection is also data-dependent. This makes the validity of conventional BH procedure suspicious and requires rigorous verification.
> - To address the difficulty arising from data-dependent selection, our main technical contribution lies in **developing a unified analytical framework**. This framework builds upon the conditional calibration framework [2], but it is not trivial due to the challenges imposed by selection in FDR control. **The number of test units **$|\hat{\mathcal{S}}_u|$** is a random variable** that can be complicatedly dependent with the p-values. Even though we can replace $|\hat{\mathcal{S}}_u|$ with $m$ for valid FDR control, it would be too loose and lead to reduced power.
> - To fix the selective randomness in analyzing FDR, we **utilize the stability property of selection rule**. Our stability condition helps mitigate the randomness in the test number which is not addressed in a single test. And dependence association between the selective p-values and the final rejection set can also be decoupled through the stability.
>
> We hope that these interpretations can ease your doubts. If you have any further questions, please feel free to ask us.
>
> [1] Jin, Ying, and Zhimei Ren. Confidence on the focal: Conformal prediction with selection-conditional coverage. arXiv, 2024.
>
> [2] William Fithian and Lihua Lei. Conditional calibration for false discovery rate control under dependence. AOS, 2022.
>
> > Weakness 2: minor errors
>
> **To W2**: Sorry for our incautiousness. Thank you for pointing out these errors. We will revise them in the future revision.
>
> > Questions: Is the idea of adapting the calibration set similar to constructing the reference set in [1]?
>
> **To Q**: Your question is very thoughtful and we would like to discuss it with you in depth.
>
> - Indeed, our adaptive strategy has the same intension as "swapped" strategy, that is to maintain the exchangeability between selected calibration set and test unit.  From a theoretical standpoint, the selective p-values constructed by both strategies are valid p-values given the selection event.
> - **However, the core objectives of our adaptive strategy and the swapped rule in  [1] and [2] are fundamentally different.** Ours focuses on conducting multiple testing over these p-values and offering theoretical guarantees. In contrast, the p-values generated by the swapped rule are utilized to construct prediction intervals with selection conditional coverage, a property specific to individual cases and not reliant on interactions between these p-values.
> - **The motivation of our adaptive strategy is directly related to weak stability**, where the selection rule satisfies $\mathbf{S}_{\mathcal{D}_c,\mathcal{D}_u}(X_i)=\mathbf{S} _{\mathcal{D}_c\cup\{Z_j\},\mathcal{D}_u\backslash\{Z_j\}}(X_i)$ for any
>  $j\in\hat{\mathcal{S}}_u$ and $i\in\mathcal{U}$. Thus, the constructed p-values are based on the selection rule of $\mathbf{S} _{\mathcal{D}_c\cup\{Z_j\},\mathcal{D}_u\backslash\{Z_j\}}$ and this is different from the swapping selection rule $\mathbf{S} _{\mathcal{D}_c\backslash\{Z_i\}\cup\{Z_j\},\mathcal{D}_u\backslash\{Z_j\}\cup\{Z_i\}}$ .
> - Moreover, **the swapping selection rules are designed to satisfy a general range of selection scenarios rather than selection rules of specific properties**. For weakly stable selection rule $\mathbf{S}$, employing the swapping selection rule$\mathbf{S}_{\mathcal{D}_c\backslash\{Z_i\}\cup\{Z_j\},\mathcal{D}_u\backslash\{Z_j\}\cup\{Z_i\}}$ can yield a selection result $\hat{S}_u^{-j}$ that differs from $\hat{S}_u$, unlike our method. This difference complicates the substitution of $\hat{S}_u^{-j}$ with $\hat{S}_u$to mitigate the randomness of the test number.
> - **From an empirical standpoint, our strategy is more computationally efficient** since for each $j\in\hat{\mathcal{S}}_u$, we only need to compute the selection rule once, while the swapping rule needs $|\mathcal{C}_0|$times.
>
> [1] Jin, Ying, and Zhimei Ren. Confidence on the focal: Conformal prediction with selection-conditional coverage. arXiv, 2024.
>
> [2] Bao, Yajie, et al. CAS: A General Algorithm for Online Selective Conformal Prediction with FCR Control. arXiv, 2024.

---

> > ### Comment · Reviewer_SifE · 2024-08-13
> >
> > Thanks to the authors for addressing my questions. I have raised my score.

---

> > > ### Author Response · Authors · 2024-08-13
> > >
> > > Thank you so much for taking the time to review our work and raising your score. If there's anything else you would like to discuss or inquire about, please don't hesitate to reach out. Once again, we appreciate your support.

---

### Official Review · Reviewer_C3MM · 2024-07-11

**Soundness:** 3
**Presentation:** 4
**Contribution:** 3
**Rating:** 6
**Confidence:** 5

**Summary:**

This paper proposes a method for multiple testing in the conformal setting that outputs the largest possible rejection set with FDR control contained with a data-dependent selection. The proposed method involves two steps:
(1) constructing selective conformal p-values (i.e., p-values solely use points in the calibration set that are chosen by the selection rule),
(2) applying the Benjamini-Hochberg (BH) procedure to the selective conformal p-values, but using a total hypothesis size of the selected set $|\widehat{\mathcal{S}}_u|$.

The method ensures FDR control only when the selection rule satisfies a form of "stability", which the authors provide several definitions for. Further, they provide several examples of practical selection rules which satisfy the stability condition (e.g., top-K, quantile), which either depend solely on the covariates in the test set, $\mathcal{D}_u$ or along with the covariates of the calibration set $\mathcal{D}_c$.

**Strengths:**

The paper provides an interesting approach to rejecting a discovery set that is a data-dependent subset of hypotheses. It's a nice combination of the techniques from [1] and [4] to solve a new selection multiple testing problem, and is similar to [3].

**Weaknesses:**

I think the main weaknesses of the paper is that it lacks comparison to two key baselines/prior art.

1) Self-consistent/compliant adjustment: Using the marginal p-values in (1), one can directly achieve FDR control under any data-dependent selection simply by taking the largest self-consistent rejection set, i.e., the largest subset $\mathcal{R}$ s.t. $p_i \leq \alpha' \mathcal{R} / K$ for each $i \in \mathcal{R} \subseteq \mathcal{S}$, where $\alpha'$ is the largest value that satisfies $\pi_0 \alpha' (1 + \log(1 / (\pi_0\alpha'))) \leq \alpha$ --- here $\pi_0 = |\mathcal{C}_0| / |\mathcal{C}|$ is the null proportion. This is a direct consequence of Theorem 3 of [5] and the PRDS property of conformal p-values from [2] referred to in your paper.

Note that this is *not* the same as your AMT baselines --- those work with the selective conformal p-values (which I presume are less powerful than the marginal conformal p-value in (1)). I think understanding the performance of your procedure to this procedure would be key to seeing how the tradeoff between the gain in power from using a smaller BH threshold vs. loss in power from using a selective conformal p-value in your method compares to a method that purely uses marginal p-values.

Post-rebuttal update: the authors ran these experiments and their method performs well. I have changed my score to an accept.

2) InfoSCOP [3]: Although InfoSCOP is cited in the paper, it is not compared against --- the FCR guarantee of InfoSCOP directly implies FDR control on a data-dependent selection set simply by making its "informative prediction set" be informative against the null hypothesis being tested (the types of sets it is informative against are precisely the types of null hypotheses you are interested in testing against). In this vein, I think a more detailed comparison of your method and the InfoSCOP method is needed (i.e., what type of selection rules are allowed for each method, and how the power differs, etc.) for this paper to be comprehensive.

I think the paper should include comprehensive comparisons to these rather significant baselines/prior art to merit publication.

**Questions:**

The assumption of having access to a calibration set and still trying to perform selective inference in a conformal setup is a bit strange. One can directly estimate a population level threshold for selecting units based (with false positive error control) on the calibration set directly, since every unit (including the test units) is drawn from the same population --- this is what allows the conformal inference to succeed in this setup. This is quite different from the outlier detection application introduced in [2], where there are units/hypotheses where the distribution is not the same as the calibration setup, and hence one cannot estimate a population level threshold, since there is no singular notion of population.

Could you elaborate on why the conformal setup makes sense here, instead of trying to directly estimate a rejection cutoff for $T(X_i)$ so the false positive rate is controlled at the population level?


- [1] Yajie Bao, Yuyang Huo, Haojie Ren, and Changliang Zou. Selective conformal inference with false coverage-statement rate control. Biometrika, 2024.
- [2] Stephen Bates, Emmanuel Candès, Lihua Lei, Yaniv Romano, and Matteo Sesia. Testing for outliers with conformal p-values. The Annals of Statistics, 2023.
- [3] Ulysse Gazin, Ruth Heller, Ariane Marandon, and Etienne Roquain. Selecting informative conformal prediction sets with false coverage
rate control. arXiv:2403.12295, 2024.
- [4] Ying Jin and Emmanuel J Candès. Selection by prediction with conformal p-values. JMLR, 2023.
- [5] Weijie Su. The FDR-Linking Theorem. arXiv:1812.08965, 2018

**Limitations:**

The limitations are well addressed.

---

> ### Author Rebuttal · Authors · 2024-08-06
>
> > Weakness 1: Comparison with self-consistent adjustment
>
> Thank you for the valuable suggestion. We have incorporated theoretical and empirical comparisons with your proposed method. And these discussions will be added in the future version of our work.
>
> - From the theoretical point of view, we have observed that the power loss associated with utilizing a selective conformal p-value is usually less than that incurred by the FDR-Linking method. To illustrate this, assume $\pi_0=0.7$ and $\alpha=0.1$ as in the simulation setting of quantile selection, then we derive $\alpha^{'}\approx0.025$.  The AMT method adjusts the marginal p-value after selection by multiplying the selection proportion $\hat{\theta}=1/0.7$. This is equivalent to employing the BH procedure on the marginal p-value with $\alpha=0.07$, which evidently yields greater power than SCA. Additionally, AMT does not make full use of the information from the selection procedure. In contrast, our proposed method uses a smaller p-value than AMT, which suggests more power increase.
> - In terms of empirical performance, as demonstrated in both cases from our paper, the SCA method suffers from a power loss, confirming our theoretical analysis.
> |  |   |  | QUAN |  | MEAN |  |
> | --- | ---  | --- | --- | --- | --- | --- |
> |  |  | FDR | Power | FDR | Power |
> | Case A | SCA  | 3.59 | 88.7 | 2.85 | 84.7 |
> |  | SPCV | 9.83 | **93.9** | 9.90 | **93.9** |
> |  | AMT  | 6.23 | 92.1 | 8.28 | 92.6 |
> | Case B | SCA  | 3.79 | 75.7 | 3.38 | 66.4 |
> |  | SPCV | 9.82 | **84.9** | 9.79 | **81.1** |
> |  | AMT  | 8.71 | 77.0 | 5.81 | 79.5 |
>
> > Weakness 2: Comparison with InfoSCOP
>
> Thank you for the constructive comments and we apologize for overlooking this important reference. As you correctly point out, the InfoSCOP involves a procedure for FDR control after selection via applying BH procedure to selective conformal p-values, which aligns closely with the fundamental approach of our work. Notably, the FDR control guarantee in InfoSCOP requires the selection rule to satisfy a specific assumption, which can be transformed into the joint-exchangeable condition in our context. For strongly stable selections, our method can be simplified and degenerate into a form similar to InfoSCOP.
>
> We would like to clarify the **following key differences** between our method and InfoSCOP:
>
> - Firstly, we provide the FDR control guarantee for general selection rules with strong stability, which is **beyond the joint-exchangeable selection**. For example, the assumption in InfoSCOP is not satisfied by the quantile selection rule based solely on test data. Thus, their theoretical results are not applicable in such cases, while our framework bridges this theoretical gap.
> -  Secondly, **our approach covers a wider range of selection rules**. For instance, when dealing with weakly stable rules, we employ conditional calibration on adaptive p-values to ensure rigorous FDR guarantees. The table below compares the performance of our approach with InfoSCOP under mean selection rule. The InfoSCOP shows reasonable empirical performance, which is similar to ours. Therefore, it is possible that InfoSCOP may still work under mean selection, making it an interesting topic for theoretical investigation, which remains unexplored in InfoSCOP. In contrast, we provide FDR control guarantee under a variety of selection scenarios.
> |  | Case A |  | Case B |  |
> | --- | --- | --- | --- | --- |
> |  | FDR | Power | FDR | Power |
> | InfoSCOP | 9.85 | 94.0 | 9.80 | 78.4 |
> | Ours | 9.86 | 93.4 | 9.80 | 78.1 |
>
> - Lastly, our approach and InfoSCOP are **designed for different goals, resulting in different analytical frameworks**. Ours is specifically designed to address the multiple testing problem across various selection rules. From the perspective of conditional calibration, our method is unified, where **the BH procedure for strongly stable selection can be seen as a special case**. As a comparison, InfoSCOP is an excellent work for selecting an informative set with FCR control, but it is **not primarily designed for multiple testing after data-dependent selection**. Their FDR guarantee is an extension of FCR control, which limits their method's applicability to different selection rules.
>
> Based on your suggestions, we will include a comprehensive comparison of our method with InfoSCOP in future version. Hope the above interpretation can ease your doubts.
>
> > Question:  Could you elaborate on why the conformal setup makes sense here?
>
> Thank you for the insightful question. We would like to make some clarifications.
>
> - Indeed,  **our method is versatile and not limited to conformal setups.** For instance, in scenarios where only null-labeled data is available, such as in outlier detection, our approach can still generate selective conformal p-values and perform the appropriate procedures to control the FDR. In such cases, we assume that the distribution of calibration data is identical to the test data because the join-exchangeable selection rule is expected to be applicable in this scenario. However, it is not necessary for quantile or mean selection rules which based on test data only.
> - Your comment on population level based selection is indeed insightful and valuable, particularly for scenarios where the goal is to directly select a subset from the original data. However, our framework offers a broader applicability beyond this. In many cases, **we are only concerned about the specific selected subgroup, and the FDR over this set should be controlled**. For example,  in brain scan experiments, researchers hope to find brain locations for specific signal with FDR control. Given that there are several encephalic regions, each should be treated separately [1]. In such cases, controlling FDR within these subsets is of concern, and a global cutoff may have no theoretical guarantee.
>
> [1] Efron, B., Simultaneous Inference: When Should Hypothesis Testing Problems Be Combined? AOAS, 2008.

---

> > ### Comment · Reviewer_C3MM · 2024-08-12
> >
> > Thank you for running experiments comparing the methods and thoroughly explaining the differences with existing work. --- I have updated my score to accept.
> >
> > As a final comment, I think it would be helpful to describe the relationship of your conditional calibration approach with the application of boosting with conditional calibration in Section 6 [1] as applied to conformal multiple testing --- can your procedure be seen as a generalization or specific case of their derandomization (or even randomized) approach to getting FDR control?
> >
> > [1] J. Lee and Z. Ren. Boosting e-BH via conditional calibration. arXiv, 2024.

---

> > > ### Author Response · Authors · 2024-08-13
> > >
> > > > Comment: Describe the relationship of your conditional calibration approach with the application of boosting with conditional calibration in Section 6 [1] as applied to conformal multiple testing. Can your procedure be seen as a generalization or specific case of their derandomization (or even randomized) approach to getting FDR control?
> > >
> > > **To C**: Thank you once again for updating the score and providing your insightful comments.
> > >
> > > - Indeed, our procedure can be viewed as a generalization for a selective scenario of their approach [1]. The conditional calibration approach with random pruning is equivalent to the e-BH procedure applied to $\{e_j/\epsilon_j:j\in\hat{\mathcal{S}}_u\}$, where $e_j=\frac{\hat{\mathcal{S}}_u\mathbb{I}(p_j\leq\frac{\alpha\hat{R}_j(\mathbf{p})}{\hat{\mathcal{S}}_u})}{\alpha\hat{R}_j(\mathbf{p})}$ and $\epsilon_j$ are independent standard uniform random variables. Additionally, our approach with deterministic pruning is equivalent to the e-BH procedure applied to $\{e_j:j\in\hat{\mathcal{S}}_u\}$.
> > > - Under our stability assumption, we can confirm that $e_j$ is a valid e-value in a manner similar to Lemma E.2 in our paper. However, exploring the property for general selection rules remains a task, as $\hat{\mathcal{S}}_u$ is a random variable correlated to $p_j$.
> > > - With this equivalence property, the boosting method [1] can be directly applied to our deterministic pruning approach by constructing the new boosting e-value to enhance power. This boosting method enhances the power of e-BH without sacrificing its FDR control or introducing additional randomness. This excellent work can significantly improve the reproducibility of results from our conditional calibration approach. We will add this discussion to the article in future versions.
> > >
> > > [1] J. Lee and Z. Ren. Boosting e-BH via conditional calibration. arXiv, 2024.

---

### Official Review · Reviewer_9oVU · 2024-07-13

**Soundness:** 3
**Presentation:** 3
**Contribution:** 2
**Rating:** 6
**Confidence:** 4

**Summary:**

This paper considers the problem of sample selection among a pre-specified group. The authors formulate this problem as a multiple testing problem and develop a procedure based on conformal inference, in which special treatment is adopted to find a specific calibration set that is exchangeable to the selected test unit. The proposed method achieves FDR control when the pre-selection rule has some strong exchangeability property.

**Strengths:**

1. The paper indeed considers an interesting problem, and the formulation makes sense.
2. The paper is well-presented and easy to follow.

**Weaknesses:**

**Technical contribution.** As introduced in the paper, the technical difficulty of sample
selection among the selected units with FDR control lies in (1) constructing valid p-values in the
presence of selection and (2) dealing with the dependency between p-values for FDR control.
The current paper mainly addresses the first problem, while the second problem is only parially solved
for quite restrictive selection rules. In fact, finding the exchangeable group/constructing
valid conformal p-values for selected units has already been quite extensively discussed in [1] and [2].
It would be helpful to clarify the technical contribution given this context.

1. Jin, Ying, and Zhimei Ren. "Confidence on the focal: Conformal prediction with selection-conditional coverage." arXiv preprint arXiv:2403.03868 (2024).
2. Bao, Yajie, et al. "CAS: A General Algorithm for Online Selective Conformal Prediction with FCR Control." arXiv preprint arXiv:2403.07728 (2024).

**Questions:**

See the "Weaknesses" section.

**Limitations:**

The paper has partially discussed its limitations.

---

> ### Author Rebuttal · Authors · 2024-08-06
>
> > Weaknesses for technical contribution: As introduced in the paper, the technical difficulty of sample selection among the selected units with FDR control lies in (1) constructing valid p-values in the presence of selection and (2) dealing with the dependency between p-values for FDR control. The current paper mainly addresses the first problem, while the second problem is only partially solved for quite restrictive selection rules. In fact, finding the exchangeable group/constructing valid conformal p-values for selected units has already been quite extensively discussed in Jin and Ren (2024) and Bao et.al (2024). It would be helpful to clarify the technical contribution given this context.
>
>  Thanks for your constructive comments on the technical contribution of our work. We acknowledge that our problem also relies on the idea of ``finding the exchangeable group/constructing valid conformal p-values for selected units'', which has been discussed in previous works [1] and [2]. However, our approach addresses a more challenging multiple testing problem. We would like to provide some clarifications to highlight the contribution of our work and explain how it differs from existing studies.
>
> - **First, our problem setup differs significantly from previous works**, which focus on constructing prediction intervals after selection.  The goal of this paper is to **conduct multiple testing after data-dependent selection** such that the final rejection set has controlled FDR in a finite sample regime. This process involves the complex interaction among the p-values and the data-dependent selection process. In contrast, [1] and [2] both proposed a swapped strategy to construct valid conformal p-values after selection and then use them to build prediction intervals with selection conditional coverage. The selection conditional coverage is an individual notion and only requires the validity of a single p-value. However, this does not account for the correlations among p-values and thus does not guarantee the validity of multiple testing procedures.
> - Second, **our main technical contribution lies in developing a unified analytical framework for handling the randomness arising from data-driven selection in the context of multiple testing**. This framework builds upon the conditional calibration framework [4], but we go beyond it by tackling the **challenges imposed by selection** in FDR control. In our approach, the number of test units is denoted as $|\hat{\mathcal{S}}_u|$, which is a random variable that can have complicated dependencies with the p-values. Although it is possible to replace $|\hat{\mathcal{S}}_u|$ with a fixed value $m$for valid FDR control, doing so would be too conservative and would result in a significant loss of power.
> - Finally, to address the data-dependent selection effects, we leverage the **stability property** of the selection rule. Through a detailed investigation of the stability properties, we demonstrate that **our procedure can have finite sample FDR control across many important selection rules**. Notably, this advantageous property of stability was not investigated in [1] and [2]. For example, under the mean selection rule, which we have confirmed to be weakly stable in our framework, [1] and [2] did not fully leverage their stability characteristics, leading them to adopt a different approach for constructing selective conformal p-values. Our approach is based on $\mathbf{S}_ {\mathcal{D}_ c\cup\{Z_ j\},\mathcal{D}_ u\backslash\{Z_j\}}$ in Section 3.3, while they are based on the swapped selection $\mathbf{S}_{\mathcal{D}_c\backslash\{Z_i\}\cup\{Z_j\},\mathcal{D}_u\backslash\{Z_j\}\cup\{Z_i\}}$. Our adaptive selective p-value is more computationally efficient as it only requires computing the selection rule once instead of $|\mathcal{C}_0|$ times for each test unit. Also, our p-value is related to the selected subset $\mathbf{S} _{\mathcal{D} _c\cup\{Z _j\},\mathcal{D} _u\backslash\{Z _j\}}$, which has a closer relation to our FDR control analysis.
>
> We hope these clarifications are helpful. If you have any further questions, please feel free to reach out.
>
> [1] Jin, Ying, and Zhimei Ren. Confidence on the focal: Conformal prediction with selection-conditional coverage. arXiv, 2024.
>
> [2] Bao, Yajie, et al. CAS: A General Algorithm for Online Selective Conformal Prediction with FCR the Control. arXiv, 2024.
>
> [3] Ying Jin and Emmanuel J Candès. Selection by prediction with conformal p-values. JMLR, 2023.
>
> [4] William Fithian and Lihua Lei. Conditional calibration for false discovery rate control under dependence. AOS, 2022.

---

> > ### Comment · Reviewer_9oVU · 2024-08-13
> >
> > Thank you for clarifying your contribution! I will raise my score to 6.

---

> > > ### Author Response · Authors · 2024-08-13
> > >
> > > Many thanks for the review and raising your score. If you have any other questions, concerns, and comments, please let us know. We would like to provide our responses and address them in the future revision. Thank You!

---

### Official Review · Reviewer_cTUv · 2024-07-16

**Soundness:** 4
**Presentation:** 3
**Contribution:** 4
**Rating:** 7
**Confidence:** 4

**Summary:**

The authors study the validity of Benjamini-Hochberg like procedure on conformal p-values computed on data selected with a particular rule. Assumptions on the rules (and examples verifying them) are mentioned and guarantees demonstrated in those cases. Experiments on synthetic and classical real data and also conducted.

**Strengths:**

The presentation is clear and progressive. The problematics are easily understood, and the difficulties of the explicitly mentioned (e.g. computing p-values on data selected by a data-dependent procedure).
The contribution is interesting for the conformal/statistics community, as there is numerous applications of the result, or at least the arguments, to complex conformal tasks.

**Weaknesses:**

I'm surprised by the lack of references to Vovk's work in particular, having studied conformal p-values and conformal testing for a long time.
Theoretically, the weakness is due mainly to the weak rule setting, although it is not a significant issue.
However, I think the experimental part is the most limiting here, being limited to a very classical testing setting in a conference where applications related to deep learning would be of interest.

**Questions:**

Could you compare or clarify the contribution with regards to the conditional calibration scheme mentioned in the article ?

**Limitations:**

The approach taken in the weak selection rule has a lower power.

---

> ### Author Rebuttal · Authors · 2024-08-06
>
> > Weakness 1: the lack of references to Vovk's work in particular, having studied conformal p-values and conformal testing for a long time.
>
> **To W1**: Thank you for the nice suggestions. We will incorporate more Vovk's work to enhance the clarity of the review. There are several literature we plan to review in the future version:
>
> [1]  Vovk, V., Gammerman, A. , and Saunders, C. . Machine-learning applications of algorithmic randomness. ICML, 1999.
>
> [2] Papadopoulos, H., Proedrou, K., Vovk, V., & Gammerman, A. . Inductive confidence machines for regression. ECML,2002
>
> [3] Vovk, V., Nouretdinov, I., & Gammerman, A. . Testing exchangeability on-line. ICML, 2003.
>
> [4] Vovk V, Lindsay D, Nouretdinov I, et al. Mondrian confidence machine. Technical Report, 2003.
>
> [5] Vovk V. Conditional validity of inductive conformal predictors. Machine Learning, 2013.
>
> > Weakness 2:  experimental part is limited to a very classical testing setting in a conference where applications related to deep learning would be of interest
>
> **To W2**: We greatly appreciate your feedback on the current experimental limitations, particularly the absence of a modern setting. Your insights have inspired us to explore the application related to deep learning.
>
> We apply our method to the field of drug discovery [1] as an initial trial. Based on the DAVIS dataset [2], we aim to identify drug-target pairs with high log binding affinity (Y). Our hypothesis is $H_{0,t}: Y_t<9.21$ with target FDR $\alpha$=10%.  To transform the structural information of protein and chemical compounds into numerical features, we attach a bidirectional recurrent neural network on top of the 1D CNN output to encode them. Subsequently, we train a small neural network with 3 layers and 5 epochs based on the encoded information $X$ and binding affinity $ Y$.
>
> Below, we present our basic results for the quantile selection rule. In this rule,  j-th sample is selected if $\hat{\mu}(X_j)$ is larger than the 35%-quantile over the predicted values in the test set. The results are outlined as follows, demonstrating that our procedure (SCPV) can control the FDR precisely.
>
> |  | FDR | Power |
> | --- | --- | --- |
> | SCPV | 9.72 | 76.67 |
> | OMT | 12.30 | 93.42 |
>
>
> Due to limited time, more experiments related to deep learning will be made in future version.
>
> [1] Ying Jin and Emmanuel J Candès. Selection by prediction with conformal p-values. JMLR, 2023.
>
> [2] Mindy I Davis, et. al. Comprehensive analysis of kinase inhibitor selectivity. Nature Biotechnology, 2011.
>
> > Question: Could you compare or clarify the contribution with regards to the conditional calibration scheme mentioned in the article?
>
> **To Q**: Your question is very meaningful. Here we make a detailed discussion about the contribution with regards to the conventional conditional calibration.
>
> The conventional conditional calibration [1] offers a flexible framework to decouple the dependence between p-values. It requires a carefully constructed quantity $\Phi_j$, such that appropriate $c_i^*$ can be identified to satisfy the condition $\mathbb{E}[\mathbb{1}\{p_j\leq c^*_i\}/|\hat{\mathcal{R}}_j|\mid \Phi_j]\leq \alpha/m$, where $p_j$ is a p-value under null, $\hat{\mathcal{R}}_j$ is a substitution of the original rejection set $\hat{\mathcal{R}}$ and $m$ is the number of test units.
>
> In our selective setting, the number of test units is $|\hat{\mathcal{S}}_u|$, which can be complicatedly dependent with both $p_j$ and $\hat{\mathcal{R}}_j$. And when analyzing the FDR, the event that j-th sample is selected is also involved.  So our our primary focus is on ensuring $\mathbb{E}\left[\frac{\mathbb{1}\{p_j\leq c^*_i,j\in\hat{\mathcal{S}}_u\}}{|\hat{\mathcal{R}}_j|}|\hat{\mathcal{S}}_u|\mid \Phi_j\right]\leq \alpha.$
> The conditional calibration framework primarily focuses on the correlation of p-values. However, a significant challenge arises because **FDR control in a selective setting involves not only individual p-values but also the selection procedure itself**. Consequently, the selective effects are unavoidable when implementing conditional calibration. To address this, we leverage the stability property of the selection rule, which allows us to effectively conduct analysis over the selected subset effectively and rigorously.
>
> [1] William Fithian and Lihua Lei. Conditional calibration for false discovery rate control under dependence. AOS, 2022.
>
> > Limitation: The approach taken in the weak selection rule has a lower power.
>
> **To L**: Thank you for pointing this out. The conditional calibration approach will lose certain power due to the pruning process.  Our simulation results shown in the Appendix are based on deterministic pruning and it lacks power indeed. However, various techniques exist, which can enhance power through randomization. Here we present the improved results of heterogeneous random pruning, whose power is nearly as powerful as the BH procedure.
>
> |  | Case A |  | Case B |  |
> | --- | --- | --- | --- | --- |
> |  | FDR | Power | FDR | Power |
> | OMT | 15.8 | 97.1 | 16.1 | 83.8 |
> | Con | 9.86 | 93.4 | 9.80 | 78.1 |
> | BH | 9.86 | 94.0 | 9.80 | 78.4 |

---

> > ### Comment · Reviewer_cTUv · 2024-08-13
> >
> > Thank you for your detailed response and addressing my concerns. I have a clearer understanding of conditional calibration as compared to your work. I moreover appreciate the additional experiments. I maintained my rating but increased my confidence in it.

---

> > > ### Author Response · Authors · 2024-08-14
> > >
> > > We greatly appreciate your efforts in reviewing our work and increasing your confidence. If there are any additional insights or suggestions you would like to share, we are eager to hear them. Thank you once again for your support.

---

### Author Rebuttal · Authors · 2024-08-06

# Response to All Reviewers
Dear reviewers, thanks for your great efforts and valuable comments on our paper! We are glad that the reviewers found our paper "considers an interesting problem", "provides an interesting approach" and "provides extensive theoretical proof". Multiple testing after data-dependent selection in the predictive setting is an important problem, and we are the first systematic investigation to provide a unified solution.
First, we would like to emphasize our contributions again.

- Conducting multiple testing after data-dependent selection in the predictive setting is crucial in numerous real-world problems and **it is barely considered in previous work**. Our work **represents the pioneering effort in tackling this problem**.
- Our proposed procedure is theoretically verified to have **FDR control for data-dependent selection** with stability which **covers a wide range of selection rules**.
- Our method can be easily **integrated with any black-box prediction model for both regression and classification settings**. Extensive numerical experiments indicate the superiority of our method.

Next, we aim to expound our main theoretical challenge.

- There are more **complex randomness** to handle such as the test number which need not be considered in a single test. Our approach **incorporates stability conditions designed for the difficulties** encountered in multiple testing. It helps us to **mitigate the randomness** in the test number and the correlation between p-values.
- The existence of **selections beyond exchangeability** **brings** **extra intricate correlations** to the selective p-values and makes it difficult to do multiple testing. Our p-values are **constructed to decouple with the stability of selection,** particularly for the p-values under weakly stable selections. This construction can effectively counteract the correlations due to the non-exchangeability.

In the rebuttals, we have clarified the unique contributions of our research by highlighting its novelty compared to existing literature and supplemented our findings with additional numerical results to strengthen the validity of our method. In addition, we also explain the technical challenges faced by our approach.

---

### Decision · Program_Chairs · 2024-09-25

**Decision:**

Accept (poster)

**Comment:**

The reviews are strong and agreement is high. The reviewers clearly appreciated
the author rebuttal and adjusted their scores accordingly. The material provided
during the rebuttal phase clearly improves the manuscript. With that in mind,
I would strongly encourage the authors to incorporate these points and suggestions into their
revision.